# From the outer space to the inner cell: deconvoluting the complexity of *Bacillus subtilis* disulfide stress responses by redox state and absolute abundance quantification of extracellular, membrane, and cytosolic proteins

Borja Ferrero-Bordera,[1] Jürgen Bartel,[1] Jan Maarten van Dijl,[2] Dörte Becher,[1] Sandra Maaß[1]

**ABSTRACT** Understanding cellular mechanisms of stress management relies on omics data as a valuable resource. However, the lack of absolute quantitative data on protein abundances remains a significant limitation, particularly when comparing protein abundances across different cell compartments. In this study, we aimed to gain deeper insights into the proteomic responses of the Gram-positive model bacterium *Bacillus subtilis* to disulfide stress. We determined proteome-wide absolute abundances, focusing on different sub-cellular locations (cytosol and membrane) as well as the extracellular medium, and combined these data with redox state determination. To quantify secreted proteins in the culture medium, we developed a simple and straightforward protocol for the absolute quantification of extracellular proteins in bacteria. We concentrated extracellular proteins, which are highly diluted in the medium, using StrataClean beads along with a set of standard proteins to determine the extent of the concentration step. The resulting data set provides new insights into protein abundances in different sub-cellular compartments and the extracellular medium, along with a comprehensive proteome-wide redox state determination. Our study offers a quantitative understanding of disulfide stress management, protein production, and secretion in *B. subtilis*.

**IMPORTANCE** Stress responses play a crucial role in bacterial survival and adaptation. The ability to quantitatively measure protein abundances and redox states in different cellular compartments and the extracellular environment is essential for understanding stress management mechanisms. In this study, we addressed the knowledge gap regarding absolute quantification of extracellular proteins and compared protein concentrations in various sub-cellular locations and in the extracellular medium under disulfide stress conditions. Our findings provide valuable insights into the protein production and secretion dynamics of *B. subtilis*, shedding light on its stress response strategies. Furthermore, the developed protocol for absolute quantification of extracellular proteins in bacteria presents a practical and efficient approach for future studies in the field. Overall, this research contributes to the quantitative understanding of stress management mechanisms and protein dynamics in *B. subtilis*, which can be used to enhance bacterial stress tolerance and protein-based biotechnological applications.

**KEYWORDS** absolute protein quantification, *Bacillus subtilis*, disulfide stress, redox proteomics, subcellular fractionation

Address correspondence to Sandra Maaß, sandra.maass@uni-greifswald.de.

The authors declare no conflict of interest.

See the funding table on p. 20.

Systems biology addresses the missing links between molecules and physiology, deciphering how biological functions arise from dynamic interactions (1). In systems biology, several levels of complexity are integrated in mathematical models, trying to

elucidate the system's behavior. To accomplish this, the discipline has been accompanied by technical developments in high-throughput omics technologies, which allowed for the acquisition of large amounts of information. This has resulted in an increase in data availability, pushing biological models' complexity and designing experiments that improved our insights on how complex living networks like cells work.

Mass spectrometry (MS)-based proteomics marked a milestone in protein research as MS application offers the possibility of having an in-depth overview of the complete set of proteins in the cell (2, 3). MS-based proteomics has bloomed in the recent decades with the development of methods for sensitive protein identification and robust and accurate protein quantification (4). Among these strategies, relative quantification, comparing protein abundances in different conditions, has expanded as a powerful tool to answer biological questions. Nonetheless, absolute data are needed for a complete insight on how organisms respond and adapt to different conditions and its mathematical modeling. For instance, protein complex stoichiometries can only be calculated based on absolute quantification data. Thus, recently, there has been an increasing interest in methods capable of absolutely quantifying protein abundances (e.g., determining the number of protein molecules per cell or per sample amount) (5–8). Such approaches, combined with other omics data, have already been proven to be potent tools to yield detailed biochemical information. For example, enzymatic parameters can be elucidated when the concentrations of metabolites associated with an enzyme and the flux it carries are known (9), or, combined with absolute abundances of mRNA, absolute protein quantification can be used for calculating the translation efficiency (10).

Most available studies providing absolute protein quantification data so far only quantified the whole-cell extract, which is naturally enriched in cytosolic proteins. However, integrated data on the cellular location of proteins and how molecules fluctuate among different cellular compartments are needed to improve available models. Yet, proteomes of different cellular localizations, such as membrane or extracellular proteins, present different technical challenges due to the nature of those compartments. A method for the absolute quantification of membrane proteins, which are characterized by their low abundance and their highly hydrophobic nature, was previously published (8). In the case of extracellular proteins, analysis is challenging as these proteins are usually highly diluted in the medium. Hence, determination of absolute protein abundances requires efficient and unbiased protein concentration as well as the integration of suitable standards to enumerate the extent of protein concentration from the culture supernatant. Many secreted proteins are well characterized and play a significant role in living organisms as they perform a wide variety of key functions such as intercell communication, competition, or nutrient acquisition (11). Still, only a few studies have aimed at providing a comprehensive view on secreted proteins, also known as the exoproteome (12).

The Gram-positive bacterium *Bacillus subtilis* is a model organism commonly used as workhorse for industrial protein production due to its ability to efficiently translocate proteins to the extracellular space, reaching the scale of gram per liter (13–15). The high secretion rates achievable for proteins produced in *B. subtilis* outcompete other commonly used workhorses as *Escherichia coli*, simplifying downstream processing and making protein production more cost effective. Nonetheless, while *B. subtilis* is a major and prolific producer of a wide range of enzymes, it is losing share in the recombinant protein market which is shifting toward the production of complex proteins such as biotherapeutics (16). These proteins normally contain post-translational modifications (PTMs) including complex disulfide-bond patterns whose production is a common limitation in *B. subtilis* strains compared to other microbial cell factories (17). Hence, deeper knowledge on how protein complexes coordinate protein translocation and extracellular folding, together with a broader understanding of how *B. subtilis* copes with oxidative stress, is needed. For that reason, methodologies that facilitate redox-state analyses and simultaneously yield comprehensive data sets on protein abundances across sub-cellular localizations are of interest. Such data sets are of high value as they

allow improvement of the strain fitness in a rational manner by identifying bottlenecks, thus expanding the application potential of *B. subtilis* as a workhorse for producing "difficult-to-express" proteins in a cost-effective manner.

In this study, disulfide stress in *B. subtilis* was selected as proof of concept. As reported earlier, addition of sub-lethal concentrations of diamide causes growth arrest without triggering cell lysis but resulting in a well-studied stress response related to thiol cross-linking in cysteines (18). The methods applied in this study aim to contribute additional data to understand how *B. subtilis* responds to disulfide stress. For this, the proteomes of the main sub-cellular localizations, i.e., cytosol and membrane, as well as the extracellular space, were absolutely quantified. To meet the technical challenges of extracellular protein analyses, a method for the absolute quantification of extracellular proteins was developed to complement the already available methods for absolute quantification of cytosolic and membrane proteins (8, 19). Absolute protein quantification was combined with the previously developed differential cysteine (DiaCys) labeling (20) to determine the thiol redox state of proteins within the different cellular and extracellular locations. The resulting data set offers valuable insights into the absolute protein abundances inside and outside the cell, which, combined with proteome-wide redox state determination, could be implemented in systems biology models for a better understanding of bacterial stresses and protein secretion, contributing to expand *B. subtilis* application for enhanced biotherapeutics production.

## RESULTS

### Absolute quantification of secreted proteins

For the first time, this study expands the current absolute protein quantification methodologies to the exoproteome in bacteria. A straightforward protocol (Fig. 1A) was developed to circumvent the challenges associated with preparation and quantification of extracellular proteins, such as the low concentration of extracellular proteins in the large volume of culture supernatant or the presence of non-proteinogenic substances in the culture medium that could potentially interfere with sample preparation. To concentrate secreted proteins from the extracellular medium in an efficient and unbiased manner, hydroxylated silica particles (StrataClean resin) were used (21, 22). The StrataClean resin was originally developed for DNA purification, but previous research showed that it is very effective in the enrichment of diluted proteins in an unbiased manner. To be able to determine the efficiency of this protein concentration and hence trace back the absolute amounts of extracellular proteins in the culture supernatant, existing methods for absolute quantification of cytosolic and membrane proteins (8, 23) were adapted. Here, a set of seven eukaryotic proteins (concentration standards) (see Materials and Methods, Table 1) was added at a known concentration to the culture supernatant prior to protein concentration. These concentration standards were later used to translate determined protein amounts to the actual quantities in the original sample. The selection of the proteins in the concentration standard mix was made to cover the physiochemical properties (molecular weight, gravy index, and isoelectric point) of the proteins in the exoproteome, thereby allowing detection and correction for potential protein species-dependent variance. Protein concentrations used in the standard mix were adapted to the expected concentration range of native proteins in the extracellular space (Table S1).

To recover the proteins bound to the StrataClean resin, gel-based electroelution was shown to be most efficient (21). To calculate absolute protein amounts based on the final shotgun MS data, a dynamic-range protein standard (UPS2) was added to the samples at this stage and subsequently in-gel digested together with the natively secreted proteins and the proteins of the concentration standard mix. Protein abundances were determined based on a well-established label-free MS approach, which, combined with UPS2, was previously shown to be best suitable for proteome-wide absolute quantification (8). For that reason, normalized intensity-based absolute quantification (riBAQ) values were used to calculate molar concentrations based on added UPS2 standards. riBAQ values are

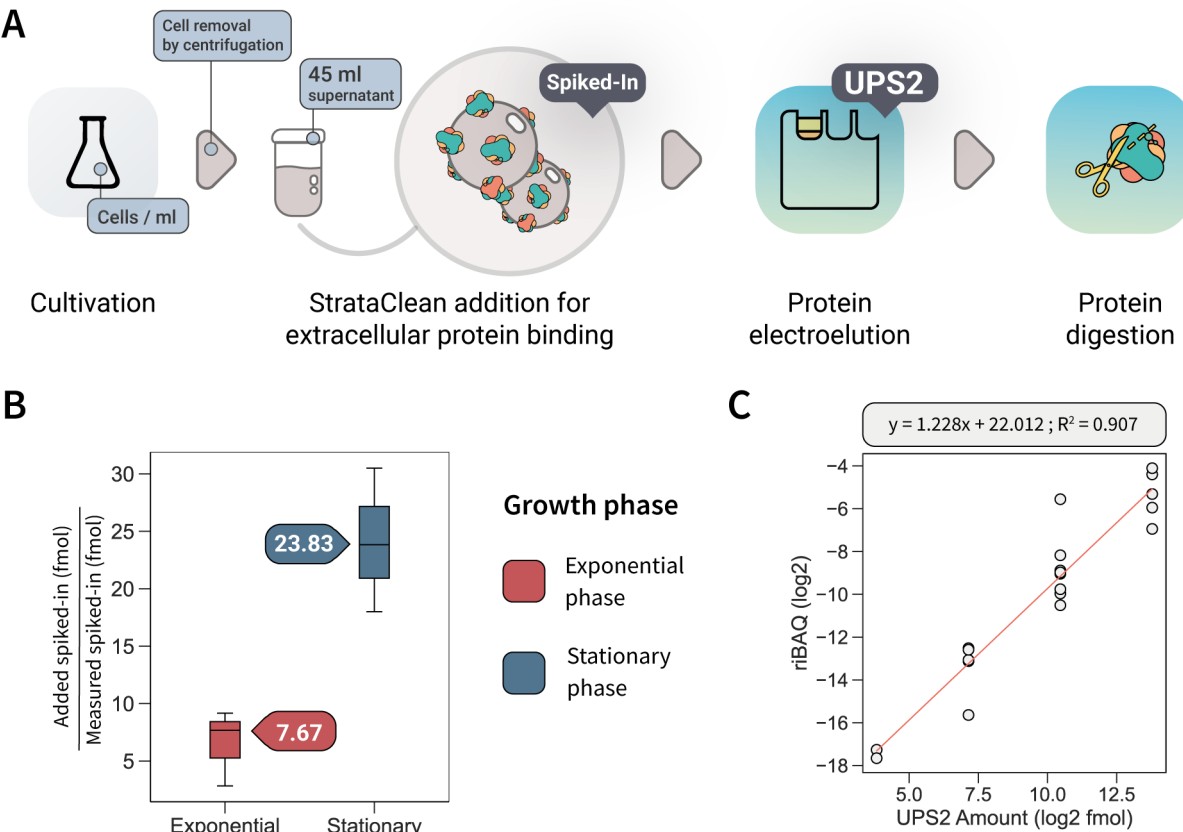

FIG 1  (A) Workflow for absolute protein quantification of extracellular proteins. Shortly, extracellular proteins present in the cell supernatant were bound on StrataClean resin together with a known amount of concentration standards (spiked-in). Then, bound proteins were electroeluted by SDS-PAGE and separated together with UPS2 standards for absolute quantification. Finally, in-gel digestion of proteins was performed prior to liquid chromatography-tandem mass spectrometry measurement. (B) Concentration factors calculated with the concentration standards in four biological replicates. (C) Calibration curve for UPS2 quantification standards.

equivalent to iBAQ values, which determine the abundance of a protein by dividing the total MS-precursor intensities by the number of theoretically observable peptides of the protein. However, in riBAQ, each protein's iBAQ value is normalized to the sum of all iBAQ values (24, 25).

To validate the method's capabilities, the workflow was tested with *B. subtilis* cells during exponential growth and in stationary phase (Fig. S1). In total, 113 proteins with predicted extracellular localization (~39% of the predicted exoproteome) were

TABLE 1  Set of spiked-in standards added on raw supernatant to quantify the concentration step on protein binding with StrataClean resin[a]

| Protein | MW[b] (kDa) | pI[b] | Gravy index |
|---|---|---|---|
| α-Lactalbumin | 14.3 | 4.4 | −0.169 |
| Glyceraldehyde-3-phosphate dehydrogenase | 37.0 | 3.5 | −0.053 |
| Alcohol dehydrogenase | 36.8 | 5.6 | 0.030 |
| Soybean trypsin inhibitor | 20.0 | 4.7 | −0.241 |
| Lysozyme | 14.30 | 11.4 | −0.150 |
| Bovine serum albumin | 69.00 | 4.8 | −0.433 |
| Carbonic anhydrase | 29.00 | 6.6 | −0.506 |

[a]The set of proteins was selected based on their molecular weight, isoelectric point, and gravy index. The candidate proteins were also checked to not exhibit shared tryptic peptides with proteins from *B. subtilis* and the applied UPS2 standards. More detailed information is provided in Table S1.
[b]pI, isoelectric point; MW, molecular weight.

quantified through the described protocol (Table S2). From these secreted proteins, 99 were quantified in exponential-phase samples, while 103 secreted proteins were quantified in stationary-phase samples with an overlap of 89 proteins.

To calculate the protein amount in the culture supernatant, the concentration factor was calculated as the ratio of the known amount of the concentration standards added before concentration of natively secreted proteins to the determined amount of standard proteins after MS analyses. A higher concentration factor was seen in the stationary phase (23.83) compared to exponential phase (7.67) (Fig. 1B), which might be explained by a stronger protein secretion and/or higher protein concentrations in the supernatant of stationary *B. subtilis* cells (22) competing with the spiked-in standards for binding to the resin.

Absolute protein abundances were calculated from MS intensities based on riBAQ values and the known amount of UPS2 standard proteins added to the samples. From the five orders of magnitude covered within the UPS2 standard, the determined linear range of quantifiable proteins covers four orders of magnitude, which agrees with earlier results on absolute quantification applying this protein standard (8). Moreover, riBAQ showed good linearity within the UPS2 standard ($r^2 = 0.907$; 50–50,000 fmol) (Fig. 1C).

The new method on absolute quantification of secreted proteins provides the possibility to expand our understanding of protein secretion in *B. subtilis* at a holistic level. If no extracellular proteolysis and a steady secretion is assumed (which would be a necessary simplification of the actual biological and chemical processes), secretion rates per cell can be calculated for proteins synthesized with signal peptides that direct these proteins into the general secretion (Sec) pathway (Table S2). Given the described assumptions, the difference in the total protein molecules in the extracellular medium between two time points and the number of cells in the culture at these time points can be used to calculate secretion rates. Thus, considering $3.02 \times 10^8$ cells/mL in exponential phase and $9.22 \times 10^8$ cells/mL for a 2-h stationary phase and using the absolute amounts of extracellular proteins determined in this study, an average secretion rate of 32 molecules/cell/min can be calculated. Moreover, considering previous data obtained under the same assay conditions, which quantified the Sec secretion system in the membrane (~56 Sec translocons/cell) (8), an estimate of 34 protein molecules/h were calculated to be secreted per individual translocon. For this calculation, a constant number of translocons over time were assumed (Table S2).

Between exponential and stationary phases, an overall accumulation of secreted molecules per cell was observed (Fig. 2A). Clustering quantified proteins into functional categories permitted us to pinpoint the cellular processes in which the most abundant and/or most accumulated proteins are involved. Among these functional categories, a prominent portion of secreted proteins can be assigned to metabolic processes (52.1% of the quantified secreted proteins in the exponential phase against 68.6% in the stationary phase). Among these proteins, enzymes like amylases or lyases involved in the extracellular degradation of specific carbon sources accumulated during the stationary phase (25.8% during exponential growth increasing to 43.6% in the stationary phase) (Fig. S2). This accumulation was accompanied by a huge increase of proteins involved in iron metabolism (81.2-fold) and more modest increases of proteins involved in cell wall synthesis (7.6-fold), motility and chemotaxis (6.4-fold), and acquisition of amino acids (5.8-fold). A decrease was observed in proteins involved in lipid metabolism (fourfold). Finally, proteins associated with cellular heterogeneity were also incremented, as was the case for proteins involved in biofilm formation (2.2-fold), with BslA being the most abundant protein (301.3 fmol/mL in the stationary phase) or for extracellular proteins that are associated with sporulation (3.1-fold) like YjfA.

The data obtained in this study also allowed elucidation of the dynamics of major components of the exoproteome when exponentially growing cells enter the stationary phase (Fig. 2B). The contact-dependent growth inhibition protein WapA and sporulation inhibitor YlqB (SivC) were kept stable in the protein-to-cell ratio in the exoproteome, while other protein amounts showed substantial increases (e.g., chitosanase Csn,

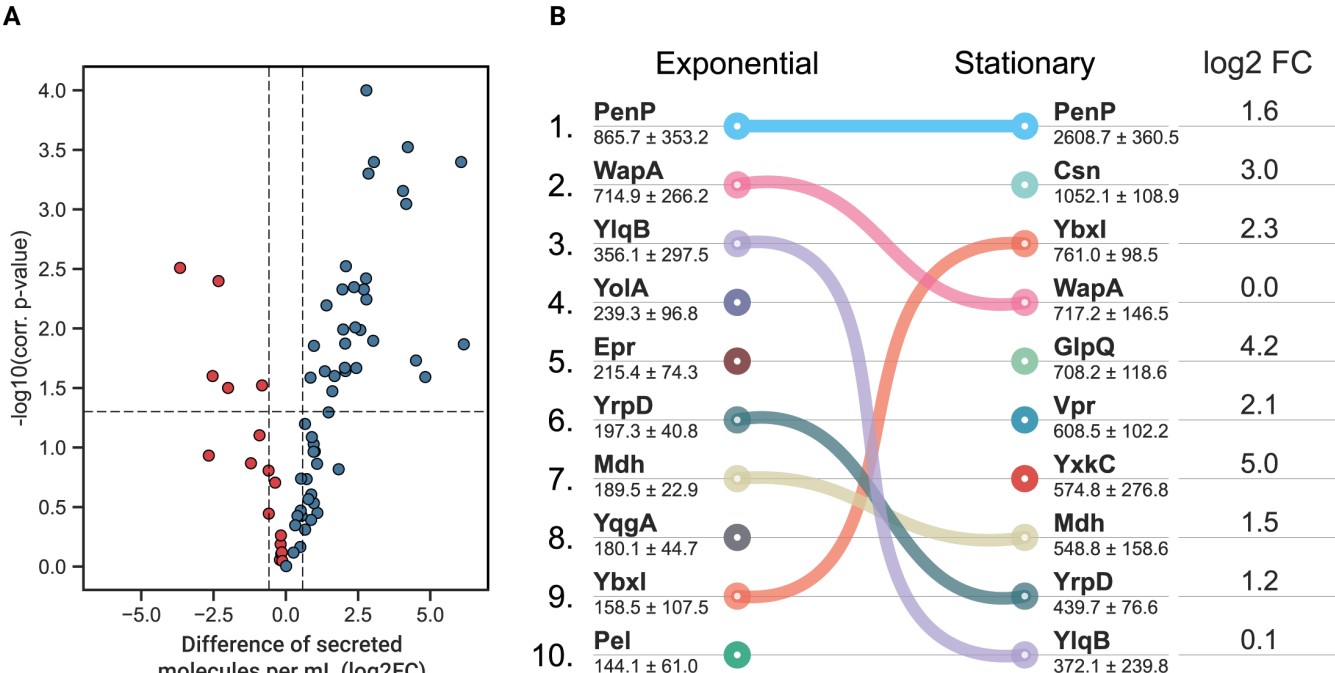

**FIG 2** (A) Difference of secreted molecules per milliliter for secreted proteins [log$_2$ fold change (FC)] between exponential and stationary phases. Blue means higher abundance in the stationary phase; red means higher abundance in the exponential phase. (B) Dynamics of the top 10 major components of the exoproteome with their concentrations indicated in femtomoles per milliliter (average ± SD). Connecting lines visualize the change in the ranking position if the protein is still among the 10 most secreted proteins. The change in protein abundance between both growth phases is summarized in the log$_2$FC column, in which fold change is calculated as the difference of log-transformed secreted proteins in femtomole per milliliter in the stationary compared to the exponential phase for the top 10 most abundant proteins in stationary phase.

glycerolphosphate diester phosphodiesterase GlpQ, extracellular serine protease Vpr, and uncharacterized protein YxkC).

## Absolute protein quantification at sub-cellular level

The method described here for absolute quantification of extracellular proteins could be easily coupled with already established protocols for absolute quantification in other sub-cellular localizations, such as the cytosol and the membrane as well as with determination of protein redox states. As proof of concept, a study on disulfide stress in *B. subtilis* was selected. As reported earlier, addition of sub-lethal concentrations of diamide causes growth arrest without triggering cell lysis but results in a well-studied stress response related to thiol cross-linking in cysteines (18). In the present study, two control conditions were established. The first control was harvested on mid-exponential phase, directly before the stress was induced (referred as "before induction"), and the second control was harvested 1 h later from an unstressed culture (referred as "control" hereafter) (Fig. S3). In parallel, *B. subtilis* was grown to mid-exponential phase and stressed with 1 mM diamide. As for the second control, also stress samples were collected 1 h after the stress onset. This condition for quantifying the effects of diamide-induced disulfide stress was selected because previous studies had shown that changes in protein abundances after 15 or 30 min of treatment with diamide are very scarce (18, 26). Both control conditions (before induction and control) can be used to show the power of integration of absolute protein quantification data obtained from different sub-cellular locations. Cell concentrations for both control conditions ($3.71 \times 10^8$ and $5.74 \times 10^8$ cells/mL for before induction and control, respectively) and the diamide stress condition ($4.92 \times 10^8$ cells/mL) were measured to calculate the numbers of copies per cell for the quantified proteins. Importantly, analyzing the control and the stress samples by absolute protein quantification at sub-cellular resolution and combined with redox

state analyses contributed to a deeper understanding of the disulfide stress effects in *B. subtilis*. Altogether, 1,609 proteins were absolutely quantified from this experiment, which includes the quantification of 228 membrane proteins and 119 secreted proteins. Details on UPS2 calibration, enrichment, and correction factors can be found in the supplemental material (Fig. S4)

Absolute proteome quantification at sub-cellular resolution permits the inference of how protein amounts fluctuate between different cell compartments (Fig. 3A). For example, the cytosolic fraction was the most densely packed sub-cellular location in the control samples. The molecule density of membrane proteins was estimated to be ~7 molecules/1,000 nm² based on the average cell surface area (0.9 µm²) (27). The data also revealed that a great portion of the determined ~2 protein molecules/µL of

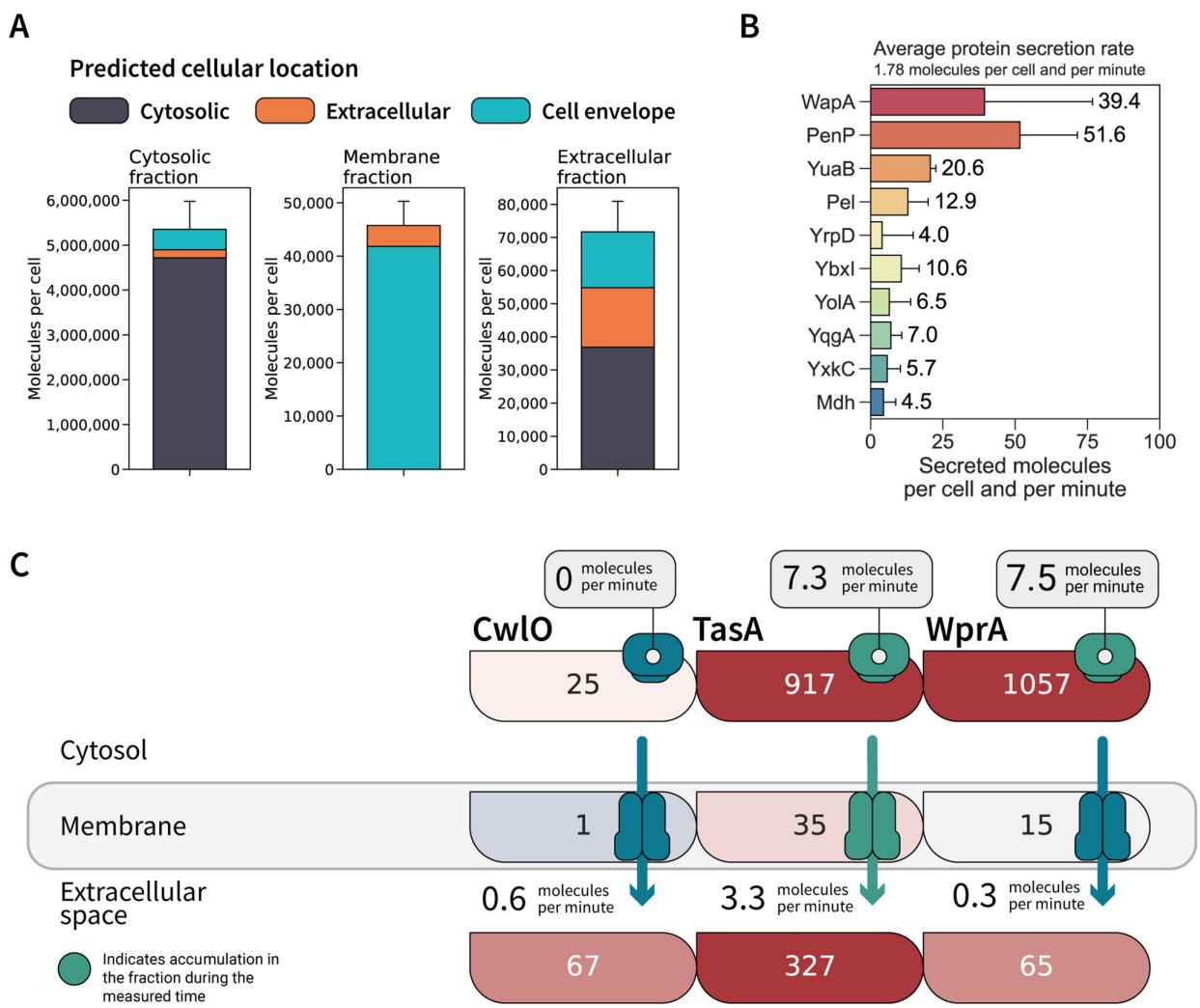

FIG 3  (A) Quantified molecules per cell assigned to proteins from each cellular location. Cytosolic proteins (115,000 ± 15,000 molecules per cell) were excluded from membrane fraction plotting to emphasize proteins assigned to the membrane proteome and translocated proteins. (B) Top 10 proteins with highest secretion rates. (C) Tracking of absolute protein abundances in molecules per cell at sub-cellular resolution for CwlO, TasA, and WprA. Protein copy numbers per cell for control samples are represented inside each box for the three different locations. Box color represents the relative abundance compared to other extracellular proteins in protein fractions from each location. Red indicates an abundance higher than the median abundance of extracellular proteins in the cellular fraction, whereas blue indicates lower abundant proteins. For the cytosol fraction, accumulation rates per minute were calculated from the difference between samples taken before induction and as control and are represented with a "ribosome-shaped" box, coded in green for active accumulation in the cytosol. For the membrane fraction, the transporter color is given in green for efficient secretion (here defined for those proteins whose secretion rate is higher than the median secretion rate quantified as 1.8 molecules/min) or in gray for standby secretion (proteins with a secretion rate lower than 1.8 molecules/min). For the extracellular fraction, secretion rates per minute are given for each protein.

culture supernatant represent cellular proteins that could result from non-conventional secretion or cell lysis. For proteins synthesized with an export signal, the abundance was even lower, with ~4 molecules/10 µL.

Individual secretion rates could be calculated for secreted proteins based on the data of both reference conditions (before induction and control). The average secretion rate of extracellular proteins was estimated to be 1.8 molecules/min and per protein. Nonetheless, our data show that much higher secretion rates can be achieved during normal growth for, e.g., PenP (51.6 secreted molecules/min) and WapA (39.4 secreted molecules/min) (Fig. 3B).

The integration of absolute protein abundances in both controls across the sub-cellular locations permitted us to draw more detail on protein secretion in *B. subtilis*. As exemplified in Fig. 3C, an accumulation of the biofilm component TasA and the protease WprA in the cytosol could be observed, while the amounts of the autolysin CwlO remained constant. However, cytosolic accumulation of extracellular proteins is not necessarily reflected in high secretion rates resulting in differential secretion efficiencies for every protein. Indeed, for WprA, which showed high intracellular levels like TasA, the secretion rate was as low as for the non-accumulated CwlO protein. The calculated active secretion of TasA is supported by data obtained from the membrane fraction, in which the number of molecules per cell is increased after 1 h.

## Absolute quantification of cellular disulfide stress

As *B. subtilis* cells were stressed with diamide in the proof-of-concept study, we used the absolute protein quantification data derived from three different sub-cellular localizations to elucidate the diamide stress response based on these data and therewith add new information to the current state of knowledge. Diamide addition is known to alter protein function by forming non-specific inter- and intra-molecular disulfide bounds between proteinogenic thiol groups. At the assayed concentration of 1 mM diamide, this disulfide formation triggers growth arrest until sufficient diamide is detoxified and normal growth can resume. In *B. subtilis*, diamide addition triggers a specific stress response known as disulfide stress with similarities to the peroxide and paraquat stress responses (18). In the present study, the concentration of 168 proteins was shown to be significantly changed (Bonferroni adjusted *P* value of <0.05, fold change of >1.5) in the cytosolic fraction. Among these differentially abundant proteins, 143 were specifically triggered by the stress induction but not by the normal growth of *B. subtilis* (comparing before induction and control). From the 143 stress-responsive proteins, 108 were increased and 35 decreased in abundance in the comparison to at least one reference condition (before induction or control, respectively). These proteins are listed in Table S3, whereas the data on all quantified proteins can be found in Table S4.

Most of the proteins detected in significantly increased amounts are related to stress response and information processing (70 out of 108), mainly protein homeostasis, while proteins with decreased abundances are mostly involved in metabolic processes (24 out of 35 depleted proteins). Our data agree with previously described responses to restore cell homeostasis under diamide stress, namely, the detoxification by accumulation of oxidoreductases together with an increased number of proteins involved in proteostasis contributing to affected protein degradation and *de novo* synthesis of proteins (18, 28).

Diamide triggers a significant accumulation of proteins involved in oxidative stress and proteostasis in the cytosol compared to both reference conditions. These proteins are summarized in Fig. 4A. Notably, the NADPH dehydrogenase NamA (YqjM) (233 molecules per cell after diamide addition) was most responsive to stress induction (24.3- to 61.8-fold). Nonetheless, the NamA abundance in the cytosol is rather low when compared to other oxidoreductases. The catalase KatA (1,078 molecules per cell), with a 3.8- to 5.7-fold induction, and the azoreductases AzoR1 (2602 molecules per cell) and AzoR2 (1,929 molecules per cell) were more abundant in the cytosol but showed more modest increases upon stress induction (2.1- to 5.7-fold).

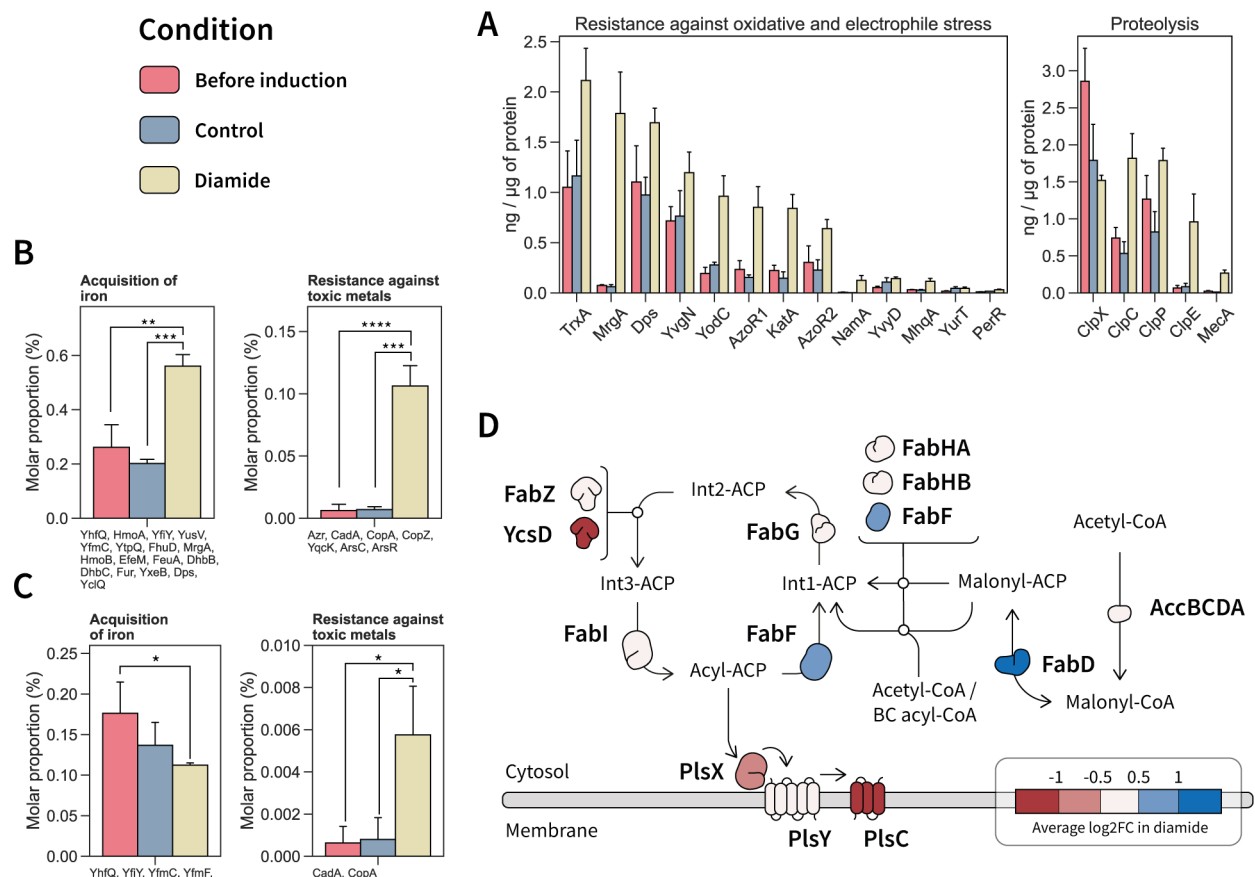

**FIG 4** Main quantified changes in the cytosol and membrane proteome in response to diamide stress. (A) Quantified cytosolic proteins involved in the oxidative stress response and protein homeostasis showing significant changes in their abundance (ng/μg of protein) under diamide stress. (B) Protein functions of which the molar proportion was changed under diamide stress in the cytosolic fraction. Quantified proteins within the functional category are listed below each plot. (C) Protein functions in the membrane of which the molar proportion was changed under diamide stress. Quantified proteins within the functional category are listed below each plot. Significance levels are expressed as * (adj. P ≤ 0.05), ** (adj. P ≤ 0.01), *** (adj. P ≤ 0.001) and **** (adj. P ≤ 0.0001). (D) Fatty acid biosynthetic pathway with quantified changes in protein abundance.

Diamide causes the accumulation and aggregation of misfolded protein in the cytosol, which needs to be resolved for the proper cell function. Consequently, the abundances of proteins within the proteostasis network were affected by diamide addition (Fig. 4A). The adaptor protein MecA (728 molecules per cell), which enables the recognition and targeting of misfolded proteins to intracellular proteases, was found among the proteins with the most increased abundance upon stress induction (11.2-fold). MecA accumulation was accompanied with an increasing abundance of the intracellular AAA protein unfoldases ClpC (1,417 molecules per cell) and ClpE (867 molecules per cell), which accumulated 2.5- to 3.4-fold and 11.6- to 14.6-fold, respectively. ClpC and ClpE form ATP-dependent Clp proteases upon association with the proteolytic sub-unit ClpP (5,784 molecules per cell), which was increased by 1.4- to 2.2-fold. Notably, in contrast to ClpC and ClpE, the levels of the AAA unfoldase ClpX (2,300 molecules per cell) decreased in the cytosol compared to the controls by 1.4- to 1.8- fold. The ClpXP complex is known to play a role in the proteolysis of the disulfide response regulator Spx (28, 29), which could only be quantified under diamide stress. In both controls, the ClpP:ClpX stoichiometry was kept at 1:1, a ratio that was altered after diamide addition (2.5:1.0).

Many biological functions arise by the accumulation of small changes in the system. Thus, by knowing the absolute abundance and functional categories of quantified

proteins, the molar proportion can be calculated for each protein category, allowing an assessment of the cellular functions that are significantly changed (Fig. 4B). An increase of 2.1- to 2.9-fold of the molar proportion in the cytosol after diamide addition was calculated for proteins involved in iron acquisition, which represent 0.6% of the molar proportion after diamide addition. Notably, in the membrane fraction (Fig. 4C), the abundance of iron acquisition proteins decreased. The depletion in the membrane was driven by a decrease in the abundance of iron transporters. Conversely, the accumulation of iron acquisition proteins in the cytosol was due to other iron proteins like the heme monooxygenase HmoA (0.17 ng/µg of protein or 933 molecules per cell, increased by 6.6- to 6.8-fold) involved in heme degradation for iron release. The iron thus acquired might be used for DNA protection through the observed increase of the ferritins MrgA (24.0- to 29.0-fold) and Dps (1.5- to 1.7-fold).

Proteins involved in resistance against toxic metals (Fig. 4B) displayed a significantly increased abundance under diamide stress in the cytosolic fraction, particularly the copper chaperone CopZ and arsenate detoxification proteins, such as ArsC, ArsR, and YqcK. This accumulation was also observed in the membrane fraction (Fig. 4C), where the abundance of the copper exporter CopA and the cadmium exporter CadA was most prominently increased. Accumulation of CadA has been linked to the release of zinc pooled by bacillithiol and zinc-containing ribosomal proteins under thiol oxidizing conditions (30).

Lipid metabolism was also affected by diamide addition, with a decrease in membrane proteins involved in phospholipid synthesis (Fig. 4D), specifically PlsX, PlsY, and PlsC. This decrease was also observed in the cytosol, with a significant decrease in PlsC (2.3- to 3.5-fold, $P$ value of <0.05 to both controls). Furthermore, diamide stress led to a depletion of the phosphatidylglycerol synthases MprF (1.7- to 2.3-fold) and PgsA (2.8-fold) in the membrane. The phosphatidylglycerol content has been suggested to affect resistance against certain compounds, and PgsA was invoked in membrane homeostasis control (31, 32). However, fatty acid synthesis did not seem to be increased, as most of the enzymes involved (FabGZI) were unchanged and the paralogous protein YcsD was depleted (Fig. 4D). Presumably, the synthesis of fatty acids was not favored as FabG and FabI need redox cofactors (33) and diamide impairs the cellular redox balance. By contrast, the abundance of the enzymes FabD and FabF increased under diamide stress. Both enzymes perform contiguous steps that lead to the synthesis of acyl-carrier protein. Notably, the FabF increase was accompanied with YuaG (FloT) accumulation (50 molecules per cell, 1.2- to 2.3-fold). Both FabF and YuaG are involved in cell envelope stress and control of membrane fluidity.

## Redox state of cellular proteins at cysteine resolution

The redox state of a protein refers to the balance of oxidized and reduced forms of its cysteine residues. The methodology implemented here, DiaCys, allows resolving of the changes in the redox state of proteins, both at cysteine and proteome-wide resolution. DiaCys is based on the quantification of cysteines differentially carbamidomethylated with light and heavy iodoacetamide, which was here applied for redox state determination of proteins in all three sub-cellular localizations. With this approach, the redox state of 8,026 cysteine-containing peptides from a total of 542 different proteins was determined. In cellular proteins located either in the cytosol or in the membrane, most of the cysteine-containing peptides were found to be natively reduced (Fig. S5). In samples without added diamide (before induction and control, respectively), ~71% of the quantified one cysteine containing peptides showed an oxidized redox state of less than 20%. These findings are in line with observations in other organisms whose redox state has been quantified at the proteome level using similar techniques (20, 34).

Unexpectedly, addition of 1 mM diamide did not cause an observable shift in the redox state of the cytosolic proteins after 1 h (74.4% of the peptides were less than 20% oxidized), emphasizing how fast the cell is able to prevent and repair cysteine damage caused by diamide. An unchanged oxidation state was also observed for peptides

which were naturally oxidized (>60% oxidized), with ~5% oxidized peptides in untreated samples and 4% 1 h after diamide addition.

Although the diamide response at the level of cysteine-containing peptides in the membrane-enriched samples was like in the cytosolic fraction, a slight increase of oxidation was observed in this proteome fraction after diamide addition compared to both controls. Indeed, only 26.9% of the total cysteine-containing peptides quantified under diamide stress were reduced (redox state <10% oxidized), while in both controls, this fraction constitutes 34.1% and 40.8% of the determined peptides, respectively.

Despite not observing large changes in the redox state of proteins in the cellular fractions, it is well known that diamide at the assayed conditions triggers a growth arrest, which is caused by thiol modification (18). To resolve the individual changes in the oxidative state of cysteines for individual proteins, the quantified cysteine redox states were statistically tested. Differentially oxidized cysteines ($P$ value of <0.01) for cytosol and membrane-enriched samples are shown in Fig. 5A. All quantified redox states are provided in Table S5.

Among the differentially oxidized cysteines in the cytosolic fraction, 6 out of 11 cysteines (54.5%) were significantly more reduced after diamide addition than in the controls. Interestingly, the redox state of the proteins containing these peptides showed a big variability between the two control conditions. This could be explained by the role of the affected cysteines as redox switches. The stress proteins Tpx and OhrB, for example, contain cysteines that showed a big difference in their oxidation state under control conditions. In both cases, the affected cysteine residues Cys60 of Tpx and Cys119 of OhrB are part of a catalytic disulfide bond, in which they undergo oxidation prior disulfide condensation (35, 36). Nevertheless, although their catalytic cysteines were differentially oxidized, the OhrB abundance was even significantly decreased under disulfide stress (2.1-fold to control, adjusted $P$ value of <0.05) while the Tpx concentration remained unchanged.

As suggested from the overall redox state determination for cellular proteins (Fig. S5), *B. subtilis*' redox state was mostly recovered 1 h after exposure to 1 mM diamide. However, it is noteworthy that most of the cysteines with significantly altered oxidation state belonged to proteins with functions in metabolic processes (Fig. 5A). Indeed, this functional group also showed a significant reduction in protein abundance (Fig. 4). These redox state changes in central functions of the cell, together with the changes in protein abundances, highlight how important redox homeostasis is for the correct functioning of a bacterial cell. As described here, minimal changes in the redox state may lead to prominent physiological effects, such as growth arrest and stress response.

By applying the herewith presented proteomic methods, changes of protein abundances and redox states within relevant cellular protein complexes can be described. Such significant redox changes were, for instance, observed in the Suf pathway, which is involved in Fe-S cluster synthesis and requires the thiol groups of Suf proteins to extract and process the sulfur atoms from free cysteine molecules (Fig. 5A). Redox changes in Suf proteins were associated with an increase in the abundances of several proteins involved in the pathway (Fig. 5B). Our data are consistent with the published stoichiometries for the *B. subtilis* SufS$_2$U$_2$ complex (37) involved in desulfuration of cysteines. Notably, the *B. subtilis* SufBCD complex was quantified with higher SufC and SufD abundances than described for the SufBC$_2$D complex of *E. coli* (38). The oxidation state of each quantified Suf protein was determined based on all quantified cysteines for each protein (Fig. 5B), showing a slight reduction in the oxidative state of proteins in the SufBCD complex, while SufS was slightly more oxidized compared to both controls.

The combination of proteome-wide redox state determination with absolute protein quantification at sub-cellular resolution provides a more comprehensive perspective on the set of mechanisms utilized by *B. subtilis* to cope with stress. Although one might speculate that high abundant proteins are more often targeted my unspecific oxidative damage as they present a higher fraction of the total protein mass in the cell, no

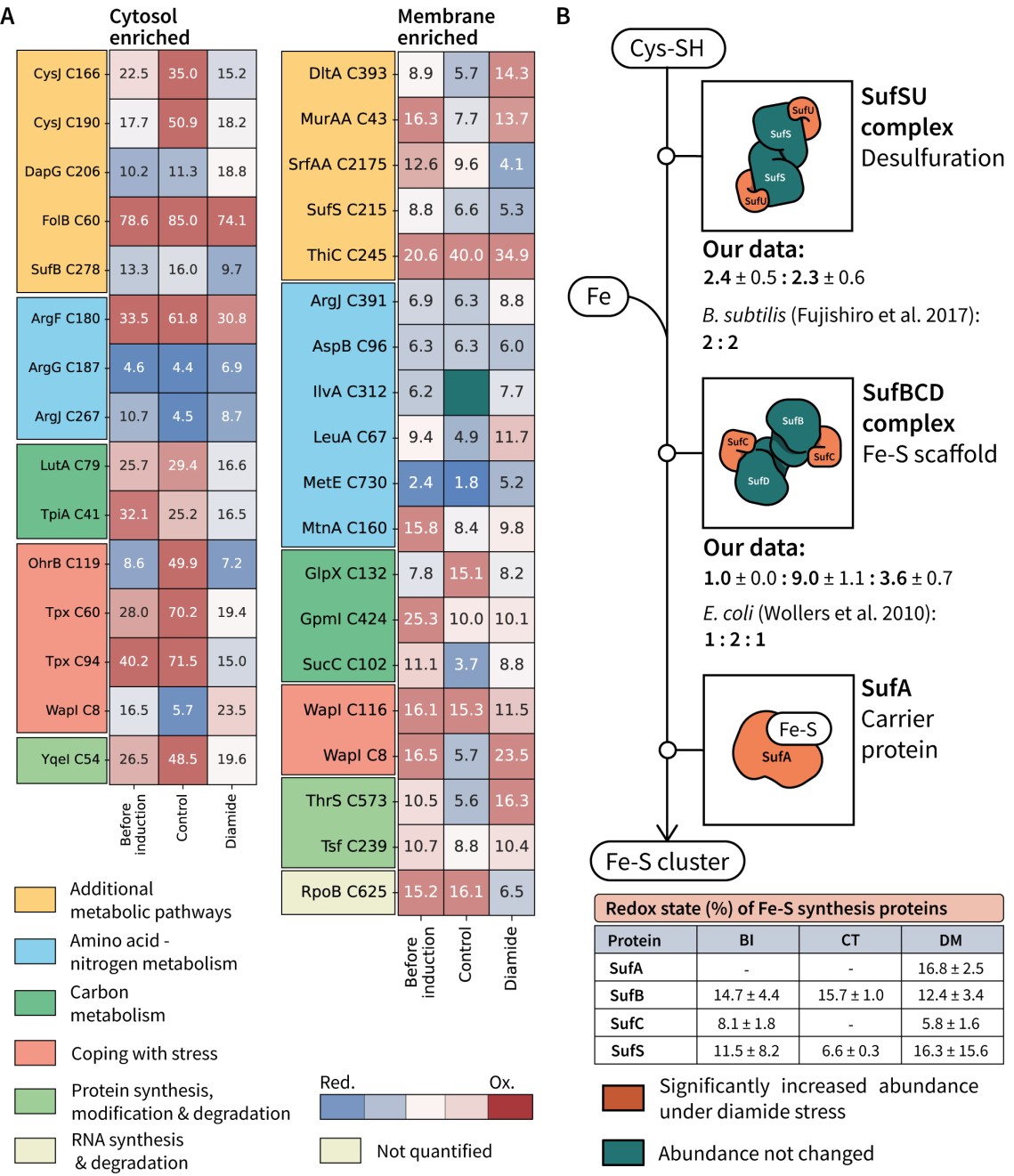

**FIG 5** (A) Significantly changed (adjusted *P* value of <0.05) cysteines in the cellular fractions obtained from cytosol and membrane-enriched samples. In-box annotation (percent oxidized) and color reflect the average redox state for the quantified cysteine. (B) Fe-S cluster synthesis pathway describing protein complex stoichiometries under disulfide stress. In the table, percentage of oxidations (average ± SD) is described for each protein identified in the Fe-S cluster synthesis pathway for the samples taken before induction (BI), as well as 1 h after addition of diamide (DM) or medium [control (CT)].

correlation was observed between protein abundance and changed oxidation rates in the cellular compartments (cytosol and membrane) after diamide addition (Fig. S6). This, together with the mostly recovered redox state of the cellular proteins (Fig. S5), highlights the speed with which *B. subtilis* can restore its cellular homeostasis following diamide-induced stress.

## Diamide effect on abundances and oxidative state of extracellular proteins

The cellular response against diamide stress has been so far mainly studied in the intracellular compartment (18, 28, 39). Although this study already added absolute protein abundance-based information and redox state analyses of cellular proteins, only little is known of the fate of proteins in the extracellular space. Nevertheless, secreted proteins are known to be key for *B. subtilis* physiology (11, 40). Thus, we offer here for the first time insights into the absolute abundance and oxidative state also for proteins obtained from the extracellular space. Our present data show a general increase in the copy numbers per protein in the culture supernatant upon diamide addition. This increase of molecules was driven mainly by a higher abundance of cytosolic proteins and lipoproteins obtained from the culture supernatant (Fig. 6A). Interestingly, molecule counts of proteins predicted to be actively secreted remained unchanged by diamide. Individual values for each quantified protein are presented in the supplemental material (Table S5).

The stress experiment revealed that diamide drives a steep oxidation of extracellular cysteines, with 86% of the quantified peptides showing an oxidation rate higher than 70% (Fig. 6C). On the other hand, the average cysteine oxidation in extracellular

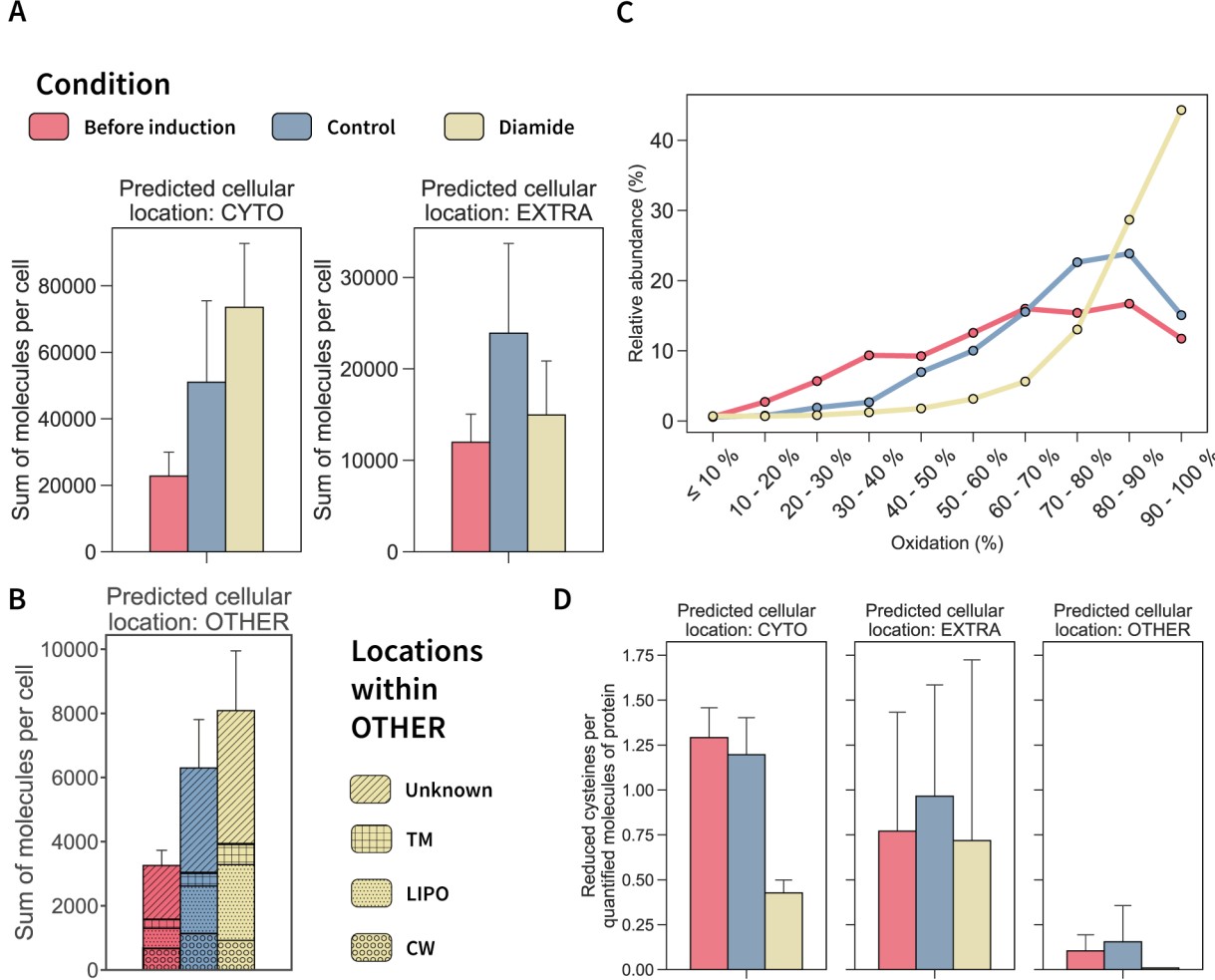

**FIG 6** Extracellular proteome response to diamide stress. (A) Total molecules per cell of quantified proteins in the supernatant for each condition grouped for cytosolic and extracellular proteins. (B) Sum of molecules per cell for proteins in the supernatant with other predicted cellular locations or unknown location. The inner pattern of each bar describes in further detail the predicted locations for proteins included in OTHER according to GP4 prediction (41). (C) Redox state of the one cysteine peptides from proteins in the extracellular space quantified for each condition. (D) Reduced cysteines per quantified molecules of protein in the supernatant for each condition grouped by their GP4-predicted cellular location.

proteins amounted already to 63.6% and 71.8%, before induction and under control conditions, respectively. These results emphasize that extracellular proteins can easily undergo significant oxidation, whereas, in contrast, redox state homeostasis in the reducing cytoplasmic environment is tightly controlled (20, 34). Additionally, our data suggest that time could play a role in the observed redox state of extracellular proteins as the amount of peptides which are more than 60% oxidized increased by 28.9% after 1 h when comparing before induction to control conditions. Furthermore, 59.7% of the peptides which were already oxidized less than 20% showed a decrease in abundance after 1 h, and peptides with a redox state between 20% and 60% oxidation decreased by 41.5%.

In contrast to observations in the cellular proteomes, in the extracellular medium higher protein abundances correlated with higher oxidation states after the addition of diamide (Fig. S6). To identify the most susceptible extracellular proteins to diamide-induced oxidation, we analyzed the redox state data in combination with the absolute protein abundances. First, we calculated the total number of cysteines per protein using sequence data and the total amount of molecules per protein. Then, the number of oxidized and reduced cysteines per protein was estimated based on the determined redox state. Oxidized and reduced cysteines were normalized by the total amount of molecules per protein. Notably, the number of reduced cysteines per molecule decreased for cytosolic proteins and lipoproteins obtained from the extracellular space (Fig. 6D), while this number remained unchanged for proteins predicted to be extracellular. This finding complements the quantified increase of molecules of cytosolic proteins and lipoproteins in the extracellular medium (Fig. 6A and B), indicating a different protein oxidation susceptibility based on the predicted cellular location of a protein.

## DISCUSSION

Systems biology represents a holistic approach to decipher complex biological systems, eventually being capable of computational integration and simulation of biological networks. For that, absolute proteomics is a powerful tool to provide necessary data and hence fill existing knowledge gaps. Determination of absolute protein abundance requires the development of accurate workflows that aim to cover all different compartments or sub-proteomes of a given cell. Sub-proteomic approaches are crucial, as protein location is key for the function of a protein or to understand processes that include protein trafficking such as cell differentiation or protein secretion. The method introduced in this study expands the quantitative proteomics toolset to a yet inaccessible and challenging part of the proteome, the extracellular proteins. Although extensive studies have already been done with extracellular proteins in all orders of life, technical challenges have made its absolute quantification at a proteome-wide level difficult. The workflow described here used StrataClean resin to quantitatively bind extracellular proteins, which are usually highly diluted in cultivation media, thereby easily recovering a standardized quantity of protein from culture supernatants without having to deal with organic solvent waste associated with other concentration methods (21). Moreover, this concentration of proteins was shown not to be biased by the physicochemical properties of the binding proteins (21). For absolute protein quantification, concentration standards were added to the sample and concentrated together with the native extracellular proteins to correct for potential binding bias and abundance effects that could be associated with the concentration step. Using intact proteins as internal standards offers the advantage of mimicking the properties of the proteins present in the sample. Similarly, Edfors and colleagues (42) applied recombinant heavy isotope-labeled proteins as standards to absolutely quantify proteins of interest in plasma samples which, like secreted proteins in bacteria, are present in highly diluted samples covering different orders of magnitude.

The presented approach for absolute quantification of extracellular proteins expands previous work on the dynamics of extracellular protein abundance. Otto and colleagues relatively quantified different sub-proteomes of *B. subtilis*; however, their extracellular

protein coverage was limited to ~15% of the predicted extracellular proteins (43). The data presented here not only expand that coverage to ~39% of the predicted secreted proteins but also provide absolute abundances for all detected extracellular proteins. Until now, to our knowledge, only a few studies absolutely quantified secreted proteins in bacteria. That was the case for the HtrA serine protease in *Campylobacter jejuni* and *Helicobacter pylori*, showing numbers around 5,000 and 9,000 secreted molecules per cell after 2 h of cultivation, respectively (44, 45). In the present study, *B. subtilis* HtrA abundance at the membrane was calculated to be 10 ± 1 molecules per cell and could not be identified in the culture supernatant. However, HtrA has been shown to be more abundant during cultivation in rich-media conditions and in strains lacking multiple extracellular proteases (11, 46). Clearly, more data from more bacterial secretomes will be necessary to compare abundances of secreted proteins and protein secretion rates.

Secretion in *B. subtilis* is described to increase during post-exponential growth (11), which agrees with the quantified protein amounts in this study. In *Streptomyces lividans*, another Gram-positive bacterium, cells also secreted more protein mass when they grew at slower growth rates. However, in this organism, total protein secretion was found to be negatively correlated with the cell density (47). The authors speculated that excessive secretion in *S. lividans* is stimulated when carbon backbones cannot be metabolically funneled toward cell growth. In Gram-negative bacteria, a positive correlation of protein amounts determined in the extracellular space and cell density has been reported (44, 45). Interestingly, when the protein amount of one of the secreted proteins, namely, HtrA, was correlated to the number of cells in the culture, constant or even decreasing secretion rates of HtrA over time were observed.

In a recent publication, the secretion rate of the recombinant staphylococcal protein IsaA was calculated to be 2.41 molecules/min in *B. subtilis* (48). These rates are in line with those quantified in the presented data, positioning recombinant protein secretion over the median secretion rate (1.78 molecules/min). Nevertheless, there seems to be room for substantial improvement as proteins like WapA show even higher secretion rates (39.41 molecules/min) at rather low cytosolic accumulation (~50 molecules per cell in both controls). Still, determining protein secretion rates per cell based on the determined absolute abundance can only be an estimate, especially if no information is available on synthesis rates and protein stability, which influence protein accumulation and depletion in the different growth phases. Even though the obtained values thus represent approximations, these data can be obtained for any secreted protein identified in the current study.

The presence of intracellular proteins in extracellular samples is noteworthy to be mentioned, as this issue constitutes one of the enigmas of extracellular proteomics. Either through non-conventional secretion or cell lysis, intracellular proteins can constitute an important fraction of the quantified extracellular protein mass (42.6% in this data set), and they can thus mask proteins that are actively secreted via the Sec system and impact their analyses (49–52). On the other hand, such "extracellular cytoplasmic proteins" have been reported for many different bacterial species, and they have been implicated in diverse functions including detoxification of the environment and adherence to host cells (50–52). The determined percentage of actively secreted extracellular proteins in the growth medium (~10% of total quantified proteins) is in line with previous studies in *B. subtilis* (~8% of total quantified proteins) (43). Nonetheless, similar approaches applied to another Gram-positive bacterium showed a recovery of ~30% actively secreted proteins (47). This difference in recovery, together with the temporal secretion differences also described in our present study, emphasizes the need to further study the mechanisms that regulate secretion and its quality control among different bacteria to unveil biological differences on secretion processes.

To date, diamide response has been studied intracellularly, describing the growth arrest as a loss of protein function caused by thiolations due to diamide (18). Nonetheless, our data suggest a more complex response to deal with disulfide stress, involving not only cytosolic proteins but also functions of the cell envelope. Changes were

observed in SigW-regulated proteins involved in fatty acid biosynthesis like FabF and in YuaG (FloT), a protein controlling membrane fluidity. In fact, previous studies in *B. subtilis* have already shown that changes in membrane fluidity could cause growth arrest highlighting the importance of membrane homeostasis (53). Similarly, the proteostasis machinery was reported to be increased when membrane fluidity was changed in yeast (54). No studies have investigated lipid composition under diamide stress to our knowledge, but the results obtained from this study suggest that a more comprehensive analyses of stress response in the context of membrane homeostasis might be beneficial.

In the present report, we describe protein abundances in molecules per cell for proteins in the two main sub-cellular localizations, cytosol and membrane, as well as the extracellular space. The integrative perspective of this comprehensive data set could help to better understand protein secretion processes. Bacilli are commonly used for recombinant protein production because of their excellent secretion capabilities, and a lot of effort has been made to improve this trait (55–58). Some of these endeavors rely on prediction tools for signal peptide selection and final sub-cellular localization (41, 59). Absolute quantification data on secreted proteins can be of great interest to expand and train computational models used for such tool development or for signal peptide screening. Indeed, the signal peptide of WapA, which was quantified to be secreted at the highest rate in our data, is commonly used in recombinant protein production (60, 61) and has been ranked as the most suitable signal peptide for recombinant protein secretion in previous studies (62).

Certainly, *B. subtilis* is a commonly used recombinant production workhorse. However, the recombinant protein market is leaning toward biotherapeutics production (16), mainly driven by monoclonal antibodies which rely on disulfide bonds for a proper folding. While *B. subtilis* has the Bdb machinery for oxidative protein folding (63), its applicability for recombinant protein production has so far not been fully explored (17). Thus, understanding and enhancing *B. subtilis* capabilities are of great interest to optimize an already available robust and reliable workhorse. A first screening of the cysteine content in proteins of different sub-cellular localizations (Fig. S7) points toward an increasing reduction of the cysteine content of proteins located in the outer layers of the cell. Here it is noteworthy that the cysteine content of proteins from Firmicutes is already relatively low, particularly in *B. subtilis* (20). In the Gram-negative model bacterium *E. coli*, oxidative protein folding occurs in the membrane and periplasmic space driven by Dsb proteins (64). This could be related to the fact that the periplasmic proteins are contained by two cellular envelope layers, the inner and outer membranes. However, in Gram-positive bacteria, secreted proteins are directly exposed to the extracellular space, which is a generally oxidizing environment. Indeed, we observed that secreted proteins undergo a rapid oxidation once they are released to the extracellular space. In fact, time-dependent effects seem to increase their oxidation. Interestingly, the predicted cysteine content per protein for extracellular proteins (~1.5 cysteines per protein) suggests that the observed oxidation would not be related to disulfide bond formation but other cysteine modifications. Nonetheless, cell wall proteins (~2 cysteines per protein) would be more likely to contain disulfide bonds. This opens the question how oxidative folding is precisely regulated and maintained for secreted proteins and how the known thiol-disulfide oxidoreductases of *B. subtilis* contribute to this process. The comprehensive data provided in this study will contribute to this understanding and build the base for further research.

## Summary and outlook

With this study, we provide a simple and straightforward methodology for the absolute quantification of secreted proteins in bacteria. The described workflow includes a low-waste protein concentration protocol together with the addition of appropriate protein standards to monitor the essential concentration step and to determine absolute protein abundances from mass spectrometric data. Altogether, these results pinpoint the secretion capabilities of *B. subtilis* and offer a comprehensive perspective on both the

secretion machinery and secreted proteins. They thus open new possibilities for optimal signal peptide screening and optimization (65). Furthermore, this new methodology can be easily coupled with absolute quantification methods for cellular sub-proteomes and with protocols for PTM analyses such as DiaCys. By application of DiaCys, this study provided deep insights on how the Gram-positive model bacterium *B. subtilis* copes with disulfide stress caused by diamide at the level of molecules per cell. Moreover, the redox state of the extracellular proteome was determined for the first time, and both methods and data can now be applied to characterize stress responses in a more holistic manner. Our work already offers a valuable data set for modeling the responses of *B. subtilis* in different cellular compartments during normal growth and after disulfide stress. Importantly, the presented methods can also be transferred to other bacteria and other physiological questions.

## MATERIALS AND METHODS

### Bacterial growth and sampling

*Bacillus subtilis* 168 *trp+* was grown in Belitsky minimal medium, and accordingly, growth was monitored by measuring the optical density at 500 nm ($OD_{500}$) (66). For benchmarking the method for absolute protein quantification of secreted proteins, three biological replicates of 100-mL glucose-limited culture were grown and harvested either during exponential growth ($OD_{500}$ 0.4) or after 2 h of stationary phase ($OD_{500}$ 1.2). Centrifugation (8,500 × *g* for 10 min at 4°C) allowed separation of cells from culture supernatant.

For disulfide stress experiments, cultivation was also performed in Belitsky minimal medium as described above with four replicates for each sample. In the exponential phase ($OD_{500}$ 0.4), diamide was added to a final concentration of 1 mM. Cultures were harvested by centrifugation before diamide addition at $OD_{500}$ 0.4 (before induction) and 1 h after adding diamide (stress) or medium (control), respectively.

### Extracellular protein concentration for absolute protein quantification

To correct secreted protein amounts detected by MS (see below) for their concentration from culture supernatants, the obtained samples were spiked with a standard. The concentration standard consisted of seven proteins from different organisms detailed in Table 1. These proteins were selected based on their different physiochemical properties such as molecular weight, pI, and hydrophobicity, considering a potential differential enrichment for different proteins. In addition, concentration standards were checked on Skyline (version 21.2, MacCoss Lab software) to not provide non-unique tryptic peptides in a *B. subtilis* background.

A stock of concentration standards was prepared based on the cited differences between selected proteins covering different amounts as explained in Table S1, aiming to preclude any possible bias caused by protein abundance in the supernatant. Standards were added based on the sample's protein amount determined using the Bradford assay to a final ratio of 1:35 (ng of standard mixture:ng of estimated sample protein).

Protein concentration from culture supernatants was performed with StrataClean resin (Agilent) as described earlier (21) with slight modifications. Briefly, the resin was primed with 37% HCl before 10.5 µL resin was incubated with the supernatants overnight at 4°C with orbital shaking to bind extracellular proteins. Resin with bound proteins was recovered by centrifugation, washed, and vacuum-dried before resuspension in 20 µL loading buffer (4% (w/v) SDS, 20% (v/v) glycerol, 40 µg/mL bromophenol blue, 125 mM Tris-HCl pH 6.8 with 10% (v/v) β-mercaptoethanol).

### Extracellular protein digestion

For MS-based absolute quantification, the universal protein standard (UPS2) (Sigma-Aldrich Merck) was used as dynamic range protein standard. UPS2 was solubilized in

loading buffer to a final concentration of 0.5 µg/µL. Samples and UPS2 were denatured at 95°C for 10 min and loaded onto precast 4-20% Tris-Glycine gels (Bio-Rad) in a ratio 1: 3.45 (µg UPS2: µL resin with bound protein) for protein electroelution at 160 V for 30 min. Afterward, gels were Coomassie stained. For diamide stress experiments, a ratio of 1:3 (µg UPS:µL resin with bound protein) was added to the first replicate of each condition.

Concentrated extracellular proteins were in-gel digested as described elsewhere (21). Briefly, gel lanes were cut in seven fractions, destained with 200 mM ammonium bicarbonate buffer in 30% (vol/vol) acetonitrile, and treated with trypsin (2 µg/µL) for protein digestion overnight. Peptides were eluted twice with water by sonication, and supernatants from the respective samples were pooled together. The peptide mixture was desalted using U-C18 Zip Tips (Merck Millipore) according to the manufacturer's instructions.

## DiaCys

DiaCys labeling was performed as described earlier (20). Shortly, in a set of four replicates per condition, natively reduced thiol groups of cellular proteins were initially labeled with 50 mM light iodoacetamide (IAA) (Sigma-Aldrich Merck) in the dark for half of the replicates. Afterward, natively oxidized thiols were reduced with Tris(2-carboxyethyl) phosphine hydrochloride (5 mM) and labeled with 50 mM stable isotope-labeled IAA ($^{13}$C2, 2-d2; 4-Da mass shift) (Sigma-Aldrich Merck). In a label-switch experiment, natively reduced thiol groups in the other half of the replicate samples were labeled with heavy IAA and natively oxidized thiols with light IAA.

For extracellular proteins, the labeling protocol was slightly modified. Proteins were bound on StrataClean for their concentration, then labeled as described above and eluted using SDS-PAGE.

## Cellular proteome preparation

Cell pellets were resuspended in Tris-EDTA (TE) buffer (10 mM Tris-HCl, 1 mM EDTA, pH 7.5) containing 50 mM IAA for labeling of natively reduced thiol groups (see above). Cells were mechanically disrupted using a ribolyzer (FastPrep24, MPBiomedicals) because this method has been shown to allow a disruption efficiency in *B. subtilis* of 99.9% (19). Cell debris and glass beads were removed by centrifugation, and protein concentration of the whole-cell protein extract was determined using the Bradford assay (Roth).

The whole-cell protein extract was considered as the representative cytosolic fraction, as it is naturally enriched in proteins contained in the cytoplasm due to their solubility in aqueous buffers. Therefore, a sample aliquot was directly used for further DiaCys labeling (see above) and digestion (see below).

Membrane proteins were enriched as published previously (8). Briefly, an initial aliquot of 770 µg whole-cell protein extract was adjusted to 1.5 mL with TE buffer (20 mM Tris-HCl, 10 mM EDTA, pH 7.5) and subjected to ultracentrifugation (109,000 × *g* at 4°C). The pellet was treated with high-salt buffer and alkaline carbonate buffer before being finally resuspended in tetraethylammonium bromide (50 mM). The sample was designated as membrane extract and subjected to further DiaCys labeling and digestion.

Cytosolic and membrane protein digestion was performed using the S-Trap protocol according to the manufacturer (Protifi) with a starting material of 7.5 µg. For shot-gun-based absolute quantification, UPS2 proteins were added in a 1:3 ratio (2.5 µg) to the first replicate of each condition. Peptide concentration was determined using the Pierce quantitative fluorometric peptide assay. For liquid chromatography-tandem mass spectrometry analysis, 2 µg of peptide mixture per biological replicate was desalted using U-C18 Zip Tips according to the manufacturer's instructions. For targeted proteomics no UPS2 but AQUA peptides were added as described previously (48) to estimate the enrichment factor of membrane proteins.

## Mass spectrometry

Peptides were separated on an Easy nLC 1000 coupled online to an Orbitrap Velos mass spectrometer (Thermo Scientific). In-house self-packed columns (i.d. 100 µm, o.d. 360 µm, length 200 mm) packed with 3-µm Dr. Maisch Reprosil C18 reversed-phase material (ReproSil-Pur 120 C18-AQ) were loaded with 18 µL of buffer A [0.1% (vol/vol) acetic acid] at a maximum pressure of 220 bar. Peptide elution was performed in a non-linear 145-min gradient with buffer B [0.1% (vol/vol) acetic acid in 95% (vol/vol) acetonitrile] for cytosol and extracellular samples at a constant flow rate of 300 nL/min. To improve the separation of hydrophobic peptides, a 180-min gradient with buffer B was used for membrane fraction samples. Precursor ion scans were obtained in the Orbitrap with a resolution of $R = 30,000$ with lock-mass correction activated. Following each MS-full scan, up to 20 tandem mass spectrometry scans were performed in a dependent manner in the linear ion trap based on precursor intensity after collision-induced dissociation fragmentation. Dynamic exclusion was enabled (exclusion size list 500, exclusion duration 20 s) with ±10 ppm exclusion window.

Membrane enrichment was assessed using AQUA peptides and targeted MS as described before (48). This method involved the use of AQUA peptides to quantitatively measure the levels of the QcrA protein in both whole-cell extract and membrane extract. For targeted MS, peptides were separated on an Easy nLC 1000 (Thermo Scientific) coupled to a triple quadrupole mass spectrometer (TSQ Vantage, Thermo Scientific). Peptide separation was performed using in-house self-packed columns packed with 3-µm Dr. Maisch Reprosil C18 reversed-phase material (see above) and applying a non-linear 100-min gradient from 1% to 99% buffer B (as above) at a constant flow rate of 300 nL/min. Q1 and Q3 selectivity was set to 0.7 Da (Full Width at Half Maximum; FWHM), and the collision gas pressure of Q2 was set at 1.2 mTorr. TSQ Vantage measurements were performed in Selected Reaction Monitoring (SRM) mode.

## Data processing

Raw data from shotgun MS was searched in MaxQuant (version 2.1.1) using the *B. subtilis* Uniprot proteome annotation (4,620 entries; UP000001570, accessed on 14 September 2021) with manually added sequences for spiked-in proteins (concentration standard) and UPS2. The following parameters were used for database search: peptide tolerance, 4.5 ppm; min fragment ions match per peptide, 2; fixed modifications, oxidation M (+15.9949), acetylation N, K (+42.0106). For disulfide stress experiment, light (+57.02146) and heavy (+61.04072) carbamidomethylations corresponding to DiaCys labeling were assigned as variable modifications. Results were filtered for a 1% false discovery rate on spectrum, peptide, and protein levels, and a minimum of two unique peptides per protein.

Skyline (version 21.2, MacCoss Lab software) was used for targeted MS raw file analysis of membrane enrichment using the transition list provided by Antelo-Varela et al. (48).

Data were processed and analyzed using Python (version 3.9). Numpy and Pandas libraries were used for data import and filtering coupled to an in-house pipeline. Normalized iBAQ (24) was used for absolute quantification of the identified proteins with a minimum of two valid values per condition for a valid quantification. The Scikit-learn library was used for calculating and assessing the UPS calibration curve and linear regression fitting. Scipy and Statsmodels packages were used for statistical analysis of the quantified proteins. Fold changes were calculated from averaged nanogram protein per microgram extract and tested through unpaired *t*-test for significance. Resulting *P* values were corrected using Bonferroni correction (alpha = 0.05). Significance was considered for fold change >1.5 and adjusted *P* value of <0.05.

To assess redox changes, only one cysteine-containing peptides were considered. Redox state was calculated by dividing the peak intensity of the oxidized form by that of the reduced form. Quantified cysteines were subjected to an unpaired *t*-test, and the

Research Article                                                                                           Microbiology Spectrum

resulting *P* values were adjusted using Bonferroni correction (alpha = 0.05). Cysteines with an adjusted *P* value of <0.05 were considered significantly changed.

For protein categorical annotation, functional categories from Subtiwiki were used. The same database was used for identifying membrane proteins, as Subtiwiki (67) is manually curated by experts in the field. For extracellular protein annotation, the GP4 signal peptide prediction tool was used (41).

68.

## ACKNOWLEDGMENTS

This work was funded by the People Programme (Marie Skłodowska-Curie Actions) of the European Union's Horizon 2020 Programme under REA grant agreement number 813979 (SECRETERS). We acknowledge support for the Article Processing Charge from the DFG and the Open Access Publication Fund of the University of Greifswald.

## AUTHOR AFFILIATIONS

[1]Department of Microbial Proteomics, University of Greifswald, Centre of Functional Genomics of Microbes, Institute of Microbiology, Greifswald, Germany
[2]Department of Medical Microbiology, University of Groningen, University Medical Center Groningen, Groningen, the Netherlands

## AUTHOR ORCIDs

Borja Ferrero-Bordera ⓘ http://orcid.org/0000-0002-6535-8607
Sandra Maaß ⓘ http://orcid.org/0000-0002-6573-1088

## FUNDING

| Funder | Grant(s) | Author(s) |
|---|---|---|
| EC \| Horizon Europe \| Excellent Science \| HORIZON EUROPE Marie Sklodowska-Curie Actions (MSCA) | 813979 | Borja Ferrero-Bordera |
| | | Jan Maarten van Dijl |
| | | Dörte Becher |

## AUTHOR CONTRIBUTIONS

Borja Ferrero-Bordera, Formal analysis, Investigation, Visualization, Writing – original draft, Writing – review and editing | Jürgen Bartel, Investigation, Methodology, Writing – review and editing | Jan Maarten van Dijl, Investigation, Supervision, Writing – review and editing | Dörte Becher, Conceptualization, Supervision, Writing – review and editing | Sandra Maaß, Conceptualization, Supervision, Writing – original draft, Writing – review and editing

## DATA AVAILABILITY

MS data have been deposited to the ProteomeXchange Consortium via the PRIDE partner repository (68) and are accessible under the identifier PXD042394.

## ADDITIONAL FILES

The following material is available online.

### Supplemental Material

**Supplemental material (Spectrum02616-23-s0001.docx).** Table S1; Figures S1 to S7.
**Table S2 (Spectrum02616-23-s0002.xlsx).** Results for exponential versus stationary phase.
**Table S3 (Spectrum02616-23-s0003.xlsx).** Significantly changed proteins after diamide addition.

**Table S4 (Spectrum02616-23-s0004.xlsx).** Absolute protein quantification data diamide experiment.

**Table S5 (Spectrum02616-23-s0005.xlsx).** Redox states of proteins.

## Open Peer Review

**PEER REVIEW HISTORY (review-history.pdf).** An accounting of the reviewer comments and feedback.

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
