## [Reviewer comments · Microbiology Spectrum]

Microbiology Spectrum

From the outer space to the inner cell: Deconvoluting the complexity of *Bacillus subtilis* disulfide stress responses by redox state and absolute abundance quantification of extracellular, membrane and cytosolic proteins

Borja Ferrero-Bordera, Jürgen Bartel, Jan Maarten van Dijl, Dörte Becher, and Sandra Maaß

Corresponding Author(s): Sandra Maaß, Universität Greifswald

Review Timeline:

Submission Date:	June 26, 2023
Editorial Decision:	October 6, 2023
Revision Received:	January 8, 2024
Accepted:	January 22, 2024

Editor: Ryan Rego

Reviewer(s): Disclosure of reviewer identity is with reference to reviewer comments included in decision letter(s). The following individuals involved in review of your submission have agreed to reveal their identity: John D Helmann (Reviewer #3)

Transaction Report:

DOI: <https://doi.org/10.1128/spectrum.02616-23>

October 6, 2023

Dr. Sandra Maaß
Universität Greifswald
Greifswald
Germany

Re: Spectrum02616-23 (From the outer space to the inner cell: Deconvoluting the complexity of *Bacillus subtilis* disulfide stress responses by redox state and absolute abundance quantification of extracellular, membrane and cytosolic proteins)

Dear Dr. Sandra Maaß:

Link Not Available

Sincerely,

Ryan Rego

Journals Department
Reviewer comments:

Reviewer #1 (Comments for the Author):

Ferrero-Bordera et al. present a new methodology for absolute quantitation of proteins in different bacterial compartments, including the extracellular space. Using their methodology, they examine the response of *Bacillus subtilis* to diamide stress at the proteome scale, as a proof-of-concept. The data is well presented, although some clarifications of the writing are required. The two concerning points are that there are a lot of known cytoplasmic proteins in the extracellular compartment, indicating significant cell lysis and complicating interpretation of findings. In addition, the timepoint assayed for the diamide stress is one hour, when the cells have mostly recovered. Shorter timepoints should be assayed to make meaningful conclusions about the *B. subtilis* response to this stress. There are minor grammatical errors throughout, so a careful proofreading is required. I have the following comments and suggestions:

Lines 93-96: This statement is disconnected from the beginning part of the paragraph. This paragraph discusses *B. subtilis* use in protein production and how it is losing ground. It should be mentioned clearly why *B. sub* is losing ground and how specifically the study of protein secretion at the proteome scale would help improve this process. For example, how is this holistic approach going to aid in *B. sub* being used to produce "difficult-to-express" proteins?

Line 122: Can you describe briefly what stratclean resin is and what are its benefits?

Line 129: It would be useful to describe the physiochemical properties here.

Lines 139-140: Describe riBAQ and iBAQ for non-experts.

Line 157. Is the r^2 obtained in line with other experiments? Typically, those values should be very close to 1, like 0.999. If those are similar to other studies, you might want to add some references.

Lines 165-166: Did you determine cellular concentrations of for your particular study? The way it is written it sounds like those values are from a reference. These numbers can vary greatly and should be calculated for your particular growth and laboratory conditions.

Lines 168-169: Are the calculated secretion rates through Sec translocons the same in log and stat phase?

Lines 172-178: You make claims about "protein secretion" consistently here and throughout the manuscript. Yet, many cytoplasmic proteins are identified in the extracellular compartment, which are likely a result of cell lysis and not secretion. It would be better to identify, and measure known or predicted secreted proteins only (Sec and others), and focus the discussion on those. Identifying sporulation proteins is likely indicative of mother cell lysis, which can explain some of the cytoplasmic proteins observed in stat phase.

Line 193: Why do you claim any "important" increase in protein abundance? You do not have any evidence that this is biologically important.

Figure S4: I might be confused, but why is the diamide treated cell enrichment so low (S4C)?

Lines 295-299: It seems odd that the cells would increase proteins to ensure resistance against toxic metals but increase uptake of additional metals. Please comment.

Line 327-331: Diamide addition likely causes redox stress, but one hour is too late to observe these changes. The experiment should be repeated on a shorter timescale since the goal of this study is to examine stress responses. Why was one hour selected? Perhaps 10, 20 and 30 minutes of exposure should be analyzed.

Line 364: MrgA is not an iron sulfur-containing protein. Reference 35 does not make such a claim. This is wrong and should be removed.

Lines 362-372: The biological significance of the changes to the SUF machinery are unclear. This section should be clarified.

Line 375: Why is the lack of correlation "interesting?" I don't find this surprising.

Lines 397-399: It has been known for a long time that the cytoplasm in bacteria is a reducing environment and is tightly regulated. Appropriate references should be found and cited.

Lines 447-454: I struggle to see the relevance of comparing *B. sub* to Gram-negative pathogens. Plus, the levels of PenP did not reach a similar abundance. It was <50% of that level and I do not see why that is biologically relevant anyway.

Lines 475-478: This section should be expanded and described in detail. It is a very important discussion.

Lines 486-487: How would this method be used to improve biotechnological methodologies? It is unclear.

Lines 488-495: This reads as random factoids. Please rewrite for clarity with more connecting and concluding sentences.

Lines 502-503: How would protein quantification data be used for training models?

Line 550 and elsewhere: Should that be OD600, which is standard for the measurement of bacterial growth?

Figure 3C: What is a "ribosome-shaped" box?

Figure 4B and C: Can the specific proteins that are being analyzed be listed.

The legends for the supplemental figures need to be as detailed as those in the main manuscript.

Minor:

Line 35: "comparedg" should be compared.

Line 51: Change "allowed to acquire huge" to "allowed for the acquisition of large amounts..."

Line 208: Remove "exemplary."

Line 212-213: Remove everything in parentheses.

Line 238-239: Change "it is tempting to use" to "we used."

Lines 319-320: Remove "namely the cytosol, membrane and extracellular space."

Line 337: Change "huge change" to "large changes..."

Line 485: I think you mean to refer to the genus "Bacillus."

Line 499: Do you mean "Bacillus species are commonly used..."

Line 567: The "range of amounts" should be explained here as well.

Line 586: What was the destaining solution and how much trypsin was used?

Lines 591-595: What were the isotopes used for heavy and light IAA and where they purchased? What concentration was used, how long were samples treated and was it in the dark?

Line 596: Should that be heavy IAA?

Line 601: Which kit was used with the FastPrep24 instrument for cell lysis?

Line 612: What is meant by resolved? Should that be reconstituted or resuspended?

Reviewer #2 (Comments for the Author):

This is an outstanding manuscript highlighting the ability to identify and quantify cytosolic, membrane-bound, and extracellular proteins in *Bacillus subtilis*. Furthermore, the authors extended the study to include the determination of cysteine redox states of these proteins following diamide exposure.

I have one comment for consideration:

1. The authors mention (Lines 327-328) that diamide did not cause an observable shift in the redox state of cytosolic proteins after 1 hour, which they attribute to efficient reduction. Could it be that 1 hour of exposure does not allow for enough diamide to reach the cytosol? This would correspond to the higher levels of oxidation of membrane proteins and extracellular proteins. It could also be that growth arrest induced by diamide is mostly due to redox modifications of membrane proteins.

Reviewer #3 (Comments for the Author):

Ferrero-Bordero et al. report a proteomics study of the disulfide stress response in *B. subtilis*. This work provides an important resource for the community of researchers interested in *B. subtilis* physiology as well as researchers interested in disulfide stress responses. The strength of the study is that it builds upon the strong prior work from this group, the proteome results are quantitative and absolute values, and the dataset includes both the membrane and exoproteome. The paper is very clear and well-written, but there are a number of places where the data could use a bit more explanation.

Major comments

1. The difference in optical density between the exponential phase and stationary phase cells is only ~10% (line 165), yet the difference in OD(500) is 3-fold (from 0.4 to 1.2; line 550). Can the authors explain this discrepancy?

2. L. 169. Can the authors reassure the reader that this estimate of number of translocons per cell is valid under their growth conditions? Since at least some proteins are translocated co-translationally, does this flux (approx. 2 min. on average for a protein to transit the membrane) make biological sense? It seems reasonable if proteins are translated at ~20 aa/second and one assumes translocation may (on average) be slower than translation, or may stall, or may involve only a subset of translocons.

3. LI 170-171. The authors refer in several places to the subset (a minority!) of the exoproteome that is predicted to transit through the Sec system. Can this annotation please be added to the data files? As presented, I did not see a column to indicate which proteins are predicted to contain signal sequences (is this the column "GP4 cellular location" in one of the SI tables?).

4. Please label the SI tables so they are numbered (as downloaded from the website it was hard to identify which was which).

5. L. 176. At this point in the text, perhaps clarify for the reader that this high fraction of metabolic enzymes is due (I think?) to the presence of many enzymes associated with cytosolic metabolic pathways in the exoproteome.

6. L. 220. Is it possible to convert these numbers into a fractional occupancy of the membrane surface? The number I have heard is close to 50%, but I would be interested in the authors' thoughts on this point.

7. L. 252. Is it possible to indicate the regulators involved? For example, which of these proteins are in the Spx regulon?

8. L. 299. The basis for induction of the *cadA* gene by diamide has been described (see Fig. 5 in PMID 25213752). Repression

of cadA requires the CzcA transcription factor that senses cellular Zn(II) levels. At the levels of DIA the authors use (1 mM), thiol oxidation triggers an increase in cytosolic Zn from both the buffered BSH pools and ribosome-associated, Zn-containing r-proteins.

9. Related mechanisms involving perturbations of cellular metal pools may account for the effects on other metal homeostasis systems, or in some cases the regulator may itself be oxidized. For example, it has been suggested that PerR may be oxidized by DIA, but in vitro studies suggest that this reaction requires high DIA levels and protein denaturants! (PMID 16766519).

10. L. 313 and Figure 4. Yes, and both are members of the SigW regulon, so a pathway for induction can be postulated. This may explain why FabF is selectively upregulated as highlighted in Fig. 4.

11. L. 471. I agree! The non-canonical secretion of cytosolic proteins to the exoproteome is a major and long-standing puzzle in proteomics. This point is, unfortunately, not as widely appreciated as it should be. I appreciate that the authors have cited the TIM review by Götz.

12. L. 899. Please define the difference between active and standby secretion.

Minor comments

II. 35, 707, 744 typos.

II. 138-139: please define abbreviations when first introduced.

I. 141. To prove (not to proof)

Staff Comments:

Preparing Revision Guidelines

Please return the manuscript within 60 days; if you cannot complete the modification within this time period, please contact me. If you do not wish to modify the manuscript and prefer to submit it to another journal, please notify me of your decision immediately so that the manuscript may be formally withdrawn from consideration by Microbiology Spectrum.

Ferrero-Bordera et al. present a new methodology for absolute quantitation of proteins in different bacterial compartments, including the extracellular space. Using their methodology, they examine the response of *Bacillus subtilis* to diamide stress at the proteome scale, as a proof-of-concept. The data is well presented, although some clarifications of the writing are required. The two concerning points are that there are a lot of known cytoplasmic proteins in the extracellular compartment, indicating significant cell lysis and complicating interpretation of findings. In addition, the timepoint assayed for the diamide stress is one hour, when the cells have mostly recovered. Shorter timepoints should be assayed to make meaningful conclusions about the *B. subtilis* response to this stress. There are minor grammatical errors throughout, so a careful proofreading is required. I have the following comments and suggestions:

Lines 93-96: This statement is disconnected from the beginning part of the paragraph. This paragraph discusses *B. subtilis* use in protein production and how it is losing ground. It should be mentioned clearly why *B. subtilis* is losing ground and how specifically the study of protein secretion at the proteome scale would help improve this process. For example, how is this holistic approach going to aid in *B. subtilis* being used to produce “difficult-to-express” proteins?

Line 122: Can you describe briefly what stratclean resin is and what are its benefits?

Line 129: It would be useful to describe the physiochemical properties here.

Lines 139-140: Describe riBAQ and iBAQ for non-experts.

Line 157. Is the r^2 obtained in line with other experiments? Typically, those values should be very close to 1, like 0.999. If those are similar to other studies, you might want to add some references.

Lines 165-166: Did you determine cellular concentrations of for your particular study? The way it is written it sounds like those values are from a reference. These numbers can vary greatly and should be calculated for your particular growth and laboratory conditions.

Lines 168-169: Are the calculated secretion rates through Sec translocons the same in log and stat phase?

Lines 172-178: You make claims about “protein secretion” consistently here and throughout the manuscript. Yet, many cytoplasmic proteins are identified in the extracellular compartment, which are likely a result of cell lysis and not secretion. It would be better to identify, and measure known or predicted secreted proteins only (Sec and others), and focus the discussion on those. Identifying sporulation proteins is likely indicative of mother cell lysis, which can explain some of the cytoplasmic proteins observed in stat phase.

Line 193: Why do you claim any “important” increase in protein abundance? You do not have any evidence that this is biologically important.

Figure S4: I might be confused, but why is the diamide treated cell enrichment so low (S4C)?

Lines 295-299: It seems odd that the cells would increase proteins to ensure resistance against toxic metals but increase uptake of additional metals. Please comment.

Line 327-331: Diamide addition likely causes redox stress, but one hour is too late to observe these changes. The experiment should be repeated on a shorter timescale since the goal of this study is to examine stress responses. Why was one hour selected? Perhaps 10, 20 and 30 minutes of exposure should be analyzed.

Line 364: MrgA is not an iron sulfur-containing protein. Reference 35 does not make such a claim. This is wrong and should be removed.

Lines 362-372: The biological significance of the changes to the SUF machinery are unclear. This section should be clarified.

Line 375: Why is the lack of correlation “interesting?” I don’t find this surprising.

Lines 397-399: It has been known for a long time that the cytoplasm in bacteria is a reducing environment and is tightly regulated. Appropriate references should be found and cited.

Lines 447-454: I struggle to see the relevance of comparing B sub to Gram-negative pathogens. Plus, the levels of PenP did not reach a similar abundance. It was <50% of that level and I do not see why that is biologically relevant anyway.

Lines 475-478: This section should be expanded and described in detail. It is a very important discussion.

Lines 486-487: How would this method be used to improve biotechnological methodologies? It is unclear.

Lines 488-495: This reads as random factoids. Please rewrite for clarity with more connecting and concluding sentences.

Lines 502-503: How would protein quantification data be used for training models?

Line 550 and elsewhere: Should that be OD₆₀₀, which is standard for the measurement of bacterial growth?

Figure 3C: What is a “ribosome-shaped” box?

Figure 4B and C: Can the specific proteins that are being analyzed be listed.

The legends for the supplemental figures need to be as detailed as those in the main manuscript.

Minor:

Line 35: “comparedg” should be compared.

Line 51: Change “allowed to acquire huge” to “allowed for the acquisition of large amounts...”

Line 208: Remove “exemplary.”

Line 212-213: Remove everything in parentheses.

Line 238-239: Change “it is tempting to use” to “we used.”

Lines 319-320: Remove “namely the cytosol, membrane and extracellular space.”

Line 337: Change “huge change” to “large changes...”

Line 485: I think you mean to refer to the genus “*Bacillus*.”

Line 499: Do you mean "*Bacillus* species are commonly used..."

Line 567: The "range of amounts" should be explained here as well.

Line 586: What was the destaining solution and how much trypsin was used?

Lines 591-595: What were the isotopes used for heavy and light IAA and where they purchased? What concentration was used, how long were samples treated and was it in the dark?

Line 596: Should that be heavy IAA?

Line 601: Which kit was used with the FastPrep24 instrument for cell lysis?

Line 612: What is meant by resolved? Should that be reconstituted or resuspended?

Journal: Microbiology Spectrum
Manuscript ID: Spectrum02616-23

Title: "From the outer space to the inner cell: Deconvoluting the complexity of *Bacillus subtilis* disulfide stress responses by redox state and absolute abundance quantification of extracellular, membrane and cytosolic proteins"

Author(s): Borja Ferrero-Bordera, Jürgen Bartel, Jan Maarten van Dijl, Dörte Becher, Sandra Maaß

Authors' response to the comments of the Reviewers

Please note that the comments by the three Reviewers were divided up and marked in black. Our responses are marked in blue. Please note also that all references to page and line numbers in our responses below relate to the 'Track Changes' version of our manuscript in which all revisions have been marked with the Track Changes tool of Word.

Author's response to the comments of Reviewer #1:

Ferrero-Bordera et al. present a new methodology for absolute quantitation of proteins in different bacterial compartments, including the extracellular space. Using their methodology, they examine the response of *Bacillus subtilis* to diamide stress at the proteome scale, as a proof-of-concept. The data is well presented, although some clarifications of the writing are required. The two concerning points are that there are a lot of known cytoplasmic proteins in the extracellular compartment, indicating significant cell lysis and complicating interpretation of findings. In addition, the timepoint assayed for the diamide stress is one hour, when the cells have mostly recovered. Shorter timepoints should be assayed to make meaningful conclusions about the *B. subtilis* response to this stress. There are minor grammatical errors throughout, so a careful proofreading is required.

Response: We thank the Reviewer for the constructive comments and suggestions, which have helped us to improve our manuscript.

I have the following comments and suggestions:

Lines 93-96: This statement is disconnected from the beginning part of the paragraph. This paragraph discusses *B. subtilis* use in protein production and how it is losing ground. It should be mentioned clearly why *B. subtilis* is losing ground and how specifically the study of protein secretion at the proteome scale would help improve this process. For example, how is this holistic approach going to aid in *B. subtilis* being used to produce "difficult-to-express" proteins?

Response: We appreciate the comment of the reviewer and have modified the text accordingly. In particular, we have more explicitly explained the benefit of comprehensive protein quantification and redox-state analyses for further improvement of *B. subtilis* strains as a chassis for biotechnological applications (lines 94-101).

Line 122: Can you describe briefly what strataclean resin is and what are its benefits?

Response: StrataClean resin is a resin that was originally developed for DNA purification, where it eliminates the need to perform phenol:chloroform extractions, as these reagents are highly toxic and combustible. The StrataClean resin is a nontoxic, noncombustible slurry of hydroxylated silica particles that can not only be used to clean up DNA, but also to concentrate dilute protein mixtures. The latter feature has been applied to enrich extracellular proteins in a highly efficient and unbiased manner

(Bonn F, Bartel J, Büttner K, Hecker M, Otto A, Becher D. 2014. Picking vanished proteins from the void: How to collect and ship/share extremely dilute proteins in a reproducible and highly efficient manner. *Anal Chem* 86:7421–7427; Otto A, Maaß S, Bonn F, Büttner K, Becher D. 2017. An Easy and Fast Protocol for Affinity Bead-Based Protein Enrichment and Storage of Proteome Samples. *Methods Enzymol* 585:1–13.) and it was therefore selected to be used in our present study. This has been explained in our revised manuscript (lines 127-129, 141-145, and 515-519).

Line 129: It would be useful to describe the physicochemical properties here.

Response: Although StrataClean resin has proven to enrich proteins from highly diluted samples in an unbiased manner (Bonn F, Bartel J, Büttner K, Hecker M, Otto A, Becher D. 2014. Picking vanished proteins from the void: How to collect and ship/share extremely dilute proteins in a reproducible and highly efficient manner. *Anal Chem* 86:7421–7427), for accurate absolute quantification of extracellular proteins this step needs to be quantified. Therefore, we searched for suitable protein standards to be used for this quantification. Hence, a set of proteins was selected based on their molecular weight, isoelectric point, and gravity index. The candidate proteins were also checked to not exhibit shared tryptic peptides with proteins from *B. subtilis* and the applied UPS2 standards. This has been described in the methods section (lines 677-681). The presently applied standard proteins resulted from this selection process and they were used to spike the samples at different concentrations to mimic the binding characteristics of proteins present in the supernatant. This information has now been added to the main text (lines 135-138) as well as to the captions of Table 1 and Supplemental Table 1.

Lines 139-140: Describe riBAQ and iBAQ for non-experts.

Response: iBAQ determines the abundance of a protein by dividing the total MS-precursor intensities by the number of theoretically observable peptides of the protein (Schwanhäusser B, Busse D, Li N, Dittmar G, Schuchhardt J, Wolf J, et al. 2011. Global quantification of mammalian gene expression control. *Nature* 473, 337–342). riBAQ is similar to iBAQ except that each protein's iBAQ value is normalized to the sum of all iBAQ values to obtain its riBAQ value (Krey JF, Wilmarth PA, Shin J-B, Klimek J, Sherman 757 NE, Jeffery ED, Choi D, David LL, Barr-Gillespie PG. 2014. Accurate label-free protein quantitation with high- and low-resolution mass spectrometers. *J Proteome Res* 13:1034–1044). We have included these explanations in the text as suggested by the reviewer (lines 147-155).

Line 157. Is the r^2 obtained in line with other experiments? Typically, those values should be very close to 1, like 0.999. If those are similar to other studies, you might want to add some references.

Response: We thank the reviewer for asking this question. Indeed, r^2 -values obtained for calibration curves from MS-based absolute quantification usually do not achieve perfect correlations as known for common biochemical methods (e.g. determination of total protein concentration) due to the complexity of the mixtures (abundances for single proteins are measured in a whole proteome background). Hence, we selected a robust method that is known to result in feasible correlation. Indeed, the r^2 -value obtained in this study is similar to many published studies using the same standards (Ahrné et al. 2013 – $r^2 = 0.87$ to 0.94 -, Sanchez et al. 2021 – $r^2 = 0.88$ to 0.91 -, Antelo-Varela et al. 2020 – $r^2 = 0.88$ to 0.94 -).

Ahrné E, Molzahn L, Glatter T, Schmidt A. 2013. Critical assessment of proteome-wide label-free absolute abundance estimation strategies. *Proteomics* 13, 2567–2578.

Sánchez BJ, Lahtvee P-J, Campbell K, Kasvandik S, Yu R, Domenzain I, Zelezniak A, Nielsen J. 2021. Benchmarking accuracy and precision of intensity-based absolute quantification of protein abundances in *Saccharomyces cerevisiae*. *Proteomics* 21, e2000093.

Antelo-Varela M, Aguilar Suárez R, Bartel J, Bernal-Cabas M, Stoberneck T, Sura T, van Dijk JM, Maaß S, Becher D. 2020. Membrane modulation of super-secreting “midiBacillus” expressing the major *Staphylococcus aureus* antigen – A mass-spectrometry-based absolute quantification approach. *Front Bioeng Biotechnol* 8.

Lines 165-166: Did you determine cellular concentrations of for your particular study? The way it is written it sounds like those values are from a reference. These numbers can vary greatly and should be calculated for your particular growth and laboratory conditions.

Response: We thank the reviewer for pointing out this critical point. In the original calculations made in this paragraph, we used the cell numbers from our previous study (Maaß, S, Wachlin, G, Bernhardt, J, Eymann, C, Fromion, V, Riedel, K, et al. 2014. Highly precise quantification of protein molecules per cell during stress and starvation responses in *Bacillus subtilis*. *Mol Cell Proteomics* 13, 2260–2276) in which the cultivation conditions were identical to our present setup (*B. subtilis* 168 Trp+, Belitzky Minimal Medium, 37 °C and same sampling points). However, we agree with the reviewer that even under the same cultivation conditions, the cell numbers may not be identical. During the revision of our manuscript, we realized that this was indeed the case. Therefore, we have recalculated all data with the actual cell counts, which we had measured during the experiment. Subsequently, all data in the text, Figures (Fig 2 and S2) and Tables (Supplemental Table 2) has been adjusted accordingly. Of note, the cells numbers were also determined in the diamide stress experiment (lines 250-252) and the corresponding results were used in each of the respective calculations.

Lines 168-169: Are the calculated secretion rates through Sec translocons the same in log and stat phase?

Response: For the calculation of secretion rates, two time-points with defined absolute protein abundances and cell numbers need to be available as secretion rates define the number of proteins translocated per time-span. Therefore, the secretion rates determined in our study are obtained by using the data available for exponentially growing and stationary cells thereby resulting in only one value. The number of Sec translocons needed for these calculations were extracted from Antelo-Varela et al. 2019 (Antelo-Varela, M, Bartel, J, Quesada-Ganuza, A, Appel, K, Bernal-Cabas, M, Sura, T, et al. (2019). Ariadne’s thread in the analytical labyrinth of membrane proteins: integration of targeted and shotgun proteomics for global absolute quantification of membrane proteins. *Anal. Chem.* 91, 11972–11980), where the same cultivation conditions were applied in the same laboratory as in our present study. We agree with the reviewer that it would be interesting to investigate whether secretion rates are different in the different growth phases (which we assume). However, with the current data from our exemplified study, this is not yet possible. Still, we are sure that such data will be available in the near future.

Lines 172-178: You make claims about "protein secretion" consistently here and throughout the manuscript. Yet, many cytoplasmic proteins are identified in the extracellular compartment, which are likely a result of cell lysis and not secretion. It would be better to identify, and measure known or predicted secreted proteins only (Sec and others), and focus the discussion on those. Identifying sporulation proteins is likely indicative of mother cell lysis, which can explain some of the cytoplasmic proteins observed in stat phase.

Response: We thank the reviewer for raising this point. We agree that cell lysis is indeed a process that imposes a challenge on exoproteome quantification, as during sample preparation proteins are enriched from culture supernatants independent of their origin. For this reason, we decided to provide the localization as predicted with the GP4 tool for each protein in the entire dataset, and we also indicated whether a protein is known to be secreted via the Sec system, as was suggested by Reviewer 3. This information is now present in Supplemental Tables 2 and 4.

Moreover, for the calculations provided in the indicated paragraph, only the abundances of proteins predicted to be secreted through the Sec pathway were included.

Line 193: Why do you claim any "important" increase in protein abundance? You do not have any evidence that this is biologically important.

Response: The reviewer is right that we do not have detailed information on the biological importance of the reported findings. In this case, we used the word "important" to point out notable changes in the protein amounts. To avoid misinterpretation by the readers of our manuscript, we changed the wording to "substantial" (line 216) and therewith hope to meet the reviewer's expectation.

Figure S4: I might be confused, but why is the diamide treated cell enrichment so low (S4C)?

Response: Membrane enrichment was quantified using targeted proteomics in all replicates of each condition based on the measurement of QcrA abundance in the non-enriched and the enriched peptide mixtures. To correct for measurement bias between shotgun and targeted proteomics, the enrichment factor was corrected with the correction factor described in Fig S4C. As the membrane enrichment relies on the different solubility of membrane-enclosed proteins, the enrichment could be susceptible to the changes in membrane composition as we have proposed in our manuscript (lines 584-605). The difference in the enrichment factor is an interesting observation that could support this hypothesis.

Lines 295-299: It seems odd that the cells would increase proteins to ensure resistance against toxic metals but increase uptake of additional metals. Please comment.

Response: We apologize that these lines in our original manuscript may have led to a misunderstanding. Indeed, proteins involved in the detoxification of toxic metals accumulate in the cytosol. In accordance with this finding an accumulation of the exporters CadA and CopA in the membrane was observed. To avoid confusion of the readers, we have now added this information in the manuscript (lines 355-363).

According to our proteome data, the acquisition of iron was reduced as deduced from the decreased abundance of iron transporters in the membrane. Furthermore, iron release into the cytosol was mostly driven by the accumulation of ferritins (MrgA, Dps) and heme proteins (HmoA) (lines 346-351). This might point to an adjusted use of the cellular iron pool in the cell under conditions of diamide stress.

Line 327-331: Diamide addition likely causes redox stress, but one hour is too late to observe these changes. The experiment should be repeated on a shorter timescale since the goal of this study is to examine stress responses. Why was one hour selected? Perhaps 10, 20 and 30 minutes of exposure should be analyzed.

Response: We agree with the reviewer that an earlier sampling might have yielded additional insights on the redox state regulation induced by diamide treatment. However, this would have been somewhat redundant with our prior work (Walgraeve J, Ferrero-Bordera B, Maaß S, Becher D, Schwerdtfeger R, van Dijl JM, Seefried M. 2023. Diamide-based screening method for the isolation of improved oxidative stress tolerance phenotypes in *Bacillus* mutant libraries. *Microbiology Spectrum*, e01608-23.), where redox responses after diamide addition were measured qualitatively, and where samples were taken after 30 minutes of diamide stress. Moreover, in the latter study and also in a study published by Leichert et al. (Leichert LIO, Scharf C, Hecker M. 2003. Global characterization of disulfide stress in *Bacillus subtilis*. *J Bacteriol* 185:1967–1975), it was shown that changes in protein abundances after 30 minutes are very scarce. For these reasons, our present study focused on the

effects of diamide one hour after stress induction. An additional major objective of our present study was to perform absolute protein quantification in relation to different subcellular locations (cytosol and membrane) as well as the extracellular medium, and to combine the data with redox state determinations. Also, for these quantitative analyses, it was important to choose a sampling time point where substantial effects could be measured, as was the case at one hour after stress induction.

Line 364: MrgA is not an iron sulfur-containing protein. Reference 35 does not make such a claim. This is wrong and should be removed.

Response: We apologize for the confusion. Our information as well as the originally cited publication have been obtained from the online Subtiwiki pathways repository, where MrgA is suggested to use Fe-S to bind to genomic DNA. It is possible that we have drawn wrong conclusions based on the information obtained from the Subtiwiki site and, therefore, we have removed the sentence as proposed by the reviewer (line 435).

Lines 362-372: The biological significance of the changes to the SUF machinery are unclear. This section should be clarified.

Response: The data on the Suf pathway were used to exemplify the power of our method to describe changes in protein abundances and redox states within relevant cellular protein complexes. We agree with the reviewer that the biological significance of this information needs to be examined in future studies. However, we believe that this is beyond the scope of our current study. To address the reviewer's comment, we have rephrased the respective text by clearly stating the purpose of the presented data (lines 431-442). We believe that this will be appreciated by the readers of our manuscript.

Line 375: Why is the lack of correlation "interesting?" I don't find this surprising.

Response: We thank the reviewer for pointing out this lack of clarity in our reasoning. Intuitively, one might think that the most abundant proteins could be more likely targets for oxidation by diamide due to their higher representation in the total protein content of a cell. As irreversibly damaged proteins (e.g. by oxidation) are usually degraded, one might also assume that there is a correlation between protein abundance and redox state changes after diamide treatment. However, no such correlation was detectable. We have now explained this more clearly in our revised manuscript (lines 445-448). In fact, the observed lack of correlation underscores the effective maintenance of redox homeostasis in the bacteria.

Lines 397-399: It has been known for a long time that the cytoplasm in bacteria is a reducing environment and is tightly regulated. Appropriate references should be found and cited.

Response: We agree with the reviewer that it is well-known that the bacterial cytoplasm is a reducing environment and that its redox state is tightly regulated. We have mentioned this in the context of our redox state determination of the cellular proteins of *B. subtilis*, where we also provided relevant references (lines 389-390). Furthermore, we have mentioned this when presenting our data on the redox states of extracellular proteins to emphasize the main difference between both locations (lines 478-480). To address the reviewer's comment, we have rephrased the respective text (lines 445-450).

Lines 447-454: I struggle to see the relevance of comparing B sub to Gram-negative pathogens. Plus, the levels of PenP did not reach a similar abundance. It was <50% of that level and I do not see why that is biologically relevant anyway.

Response: To date, the absolute quantification of exoproteomes has only rarely been performed for bacteria and such data is very scarce. We only found abundance data for a few proteins like HtrA in the studies that we cited. Considering the differences, we used these results to compare the ranges of our quantification to the already known estimates. To address the reviewer's comment, we added a sentence on the value of more comprehensive datasets, which are urgently needed and may be generated in future studies (lines 538-540).

We appreciate the reviewer's point regarding PenP and have deleted the corresponding sentence (line 540).

Lines 475-478: This section should be expanded and described in detail. It is a very important discussion.

Response: We thank the reviewer for the comment and the interest in this aspect of our study. Accordingly, we have extended the paragraph as suggested (lines 541-558).

Lines 486-487: How would this method be used to improve biotechnological methodologies? It is unclear.

Response: The biotechnological application potential of Gram-positive bacterial cell factories like *B. subtilis* depends on their secretion capabilities in terms of the yields of secreted proteins and their biological activity. Comprehensive quantitative information on protein secretion will help to identify bottlenecks in the production and secretion of the proteins of interest. With this knowledge, production processes can be further optimized by strain engineering or adaptation of the fermentation processes through alterations in e.g. the nutrient supply, or physical parameters. We have addressed these aspects in detail in the Discussion section of our revised manuscript (lines 608 ff.). Hence, we deleted the original sentence mentioned by the reviewer (line 585) to avoid redundancies.

Lines 488-495: This reads as random factoids. Please rewrite for clarity with more connecting and concluding sentences.

Response: We thank the reviewer for the suggestion. The paragraph has been rewritten (lines 586-607), and we trust that it is now easier to follow for the reader.

Lines 502-503: How would protein quantification data be used for training models?

Response: Quantitative proteomics, and especially methods for absolute protein quantification, provide key information to build and train computational models that help to predict resource allocation and therewith also contribute to our understanding of protein trafficking between cellular locations (Bulović A, Fischer S, Dinh M, Golib F, Liebermeister W, Poirier C, et al. 2019. Automated generation of bacterial resource allocation models. *Metab Eng* 55, 12–22, 1; Zeng H, Rohani R, Huang WE and Yang, A. 2021. Understanding and mathematical modelling of cellular resource allocation in microorganisms: a comparative synthesis. *BMC Bioinformatics* 22, 467). Suitable computational models might thus help us to identifying bottlenecks in protein secretion (e.g. by predicting suitable signal peptides), and they can thereby help us to improve recombinant protein production production by strain engineering or alterations in the fermentation process. We have revised the wording of the respective text and trust that it is clearer now (line 615).

Line 550 and elsewhere: Should that be OD600, which is standard for the measurement of bacterial growth?

Response: The reviewer is right that measurements of optical density to monitor bacterial growth are usually carried out at 600 nm. However, growth in Belitsky Minimal Medium, as performed in this study, is normally assessed at OD₅₀₀ due to its transparent color (Mars RAT, Mendonça K, Denham EL and van Dijk, JM. 2015. The reduction in small ribosomal subunit abundance in ethanol-stressed cells of *Bacillus subtilis* is mediated by a SigB-dependent antisense RNA. *Biochim Biophys Acta* 1853, 2553–2559; Wenzel M, Kohl B, Münch D, Raatschen N, Albada HB, Hamoen L, et al. 2012. Proteomic response of *Bacillus subtilis* to lantibiotics reflects differences in interaction with the cytoplasmic membrane. *Antimicrobial Agents and Chemotherapy* 56, 5749–5757; Chi BK, Gronau K, Mäder U, Hessling B, Becher D and Antelmann H. 2011. S-Bacillithiolation protects against hypochlorite stress in *Bacillus subtilis* as revealed by transcriptomics and redox proteomics. *Mol Cell Proteomics* 10, M111.009506.). This has not been mentioned in lines 666-667.

Figure 3C: What is a "ribosome-shaped" box?

Response: In Figure 3C, we have represented the cytosolic protein accumulation rates per minute within these small boxes that describe the scenarios observed for CwIO, TasA, and WprA. To distinguish these boxes, we used a simplified icon with the shape of a ribosome, which is commonly drawn with its two subunits, the large and the small subunit. We have described this in the Figure legend (lines 1027-1030) and trust that it will now be clear for the reader what we mean.

Figure 4B and C: Can the specific proteins that are being analyzed be listed.

Response: We thank the reviewer for this suggestion. We have revised Figure 4 accordingly, and it now lists the quantified proteins assigned to each of the different functional groups under each plot.

The legends for the supplemental figures need to be as detailed as those in the main manuscript.

Response: We revised the figure legends in the Supplemental Materials, and they now have the same format as the figure legends in the main manuscript.

Minor:

Line 35: "comparedg" should be compared.

Line 51: Change "allowed to acquire huge" to "allowed for the acquisition of large amounts..."

Line 208: Remove "exemplary."

Line 212-213: Remove everything in parentheses.

Line 238-239: Change "it is tempting to use" to "we used."

Lines 319-320: Remove "namely the cytosol, membrane and extracellular space."

Line 337: Change "huge change" to "large changes..."

Response: The suggested changes have been implemented in the text.

Line 485: I think you mean to refer to the genus "Bacillus."

Line 499: Do you mean "Bacillus species are commonly used..."

Line 567: The "range of amounts" should be explained here as well.

Line 586: What was the destaining solution and how much trypsin was used?

Lines 591-595: What were the isotopes used for heavy and light IAA and where they purchased? What concentration was used, how long were samples treated and was it in the dark?

Response: The missing information has been included in our revised manuscript, and possible ambiguities have been removed by rephrasing the text where applicable.

Line 596: Should that be heavy IAA?

Line 612: What is meant by resolved? Should that be reconstituted or resuspended

Response: The respective text has been revised to address the reviewer's questions.

Line 601: Which kit was used with the FastPrep24 instrument for cell lysis? ?

Response: For mechanical disruption of bacterial cells with a ribolyzer (e.g. FastPrep24 from MPBiomedicals) no kit was used. Instead, we used glass beads (diameter: 0.10 - 0.11 mm) that were added to the cell suspension to break the bacterial cell envelope.

Author's response to the comments of Reviewer #2:

This is an outstanding manuscript highlighting the ability to identify and quantify cytosolic, membrane-bound, and extracellular proteins in *Bacillus subtilis*. Furthermore, the authors extended the study to include the determination of cysteine redox states of these proteins following diamide exposure.

Response: We thank the Reviewer for the positive feedback on our manuscript.

I have one comment for consideration:

1. The authors mention (Lines 327-328) that diamide did not cause an observable shift in the redox state of cytosolic proteins after 1 hour, which they attribute to efficient reduction. Could it be that 1 hour of exposure does not allow for enough diamide to reach the cytosol? This would correspond to the higher levels of oxidation of membrane proteins and extracellular proteins. It could also be that growth arrest induced by diamide is mostly due to redox modifications of membrane proteins.

Response: We appreciate the comment of the reviewer. Indeed, it would be very interesting to study how the onset of the diamide-imposed stress occurs, which are the first proteins affected by the addition of diamide, which subcellular location is first affected and what are the diamide concentrations at each subcellular location over time. Our data shows that one hour after the diamide addition, the redox state of membrane proteins is more affected than that of cytosolic proteins. In a previous study (Walgraeve J, Ferrero-Bordera B, Maaß S, Becher D, Schwerdtfeger R, van Dijl JM, Seefried M. 2023. Diamide-based screening method for the isolation of improved oxidative stress tolerance phenotypes in *Bacillus* mutant libraries. *Microbiology Spectrum*, e01608-23), for which the objective was to characterize the qualitative redox responses to diamide, we observed already 30 minutes after the addition of diamide a shift in the redox states of cellular proteins (i.e. cytosolic and membrane proteins). Similarly, Sievers et al. (Sievers S, Dittmann S, Jordt T, Otto A, Hochgräfe F, Riedel K. 2018. Comprehensive redox profiling of the thiol proteome of *Clostridium difficile*. *Molecular & Cellular Proteomics* 17:1035–1046) reported a shift in the redox status of the proteome of *C. difficile* treated with 2 mM diamide already after 15 minutes. However, the main objective of our present study was to perform absolute protein quantification at different locations inside and outside of the cell. Importantly, the analysis of the redox state of proteins was used as an example to showcase the value of our method for absolute protein quantification under different experimental conditions, and to quantify the proteome changes upon diamide stress. Since it was already known from previous studies that changes in protein abundances after 15 or 30 minutes of treatment with diamide are very scarce

(Leichert LIO, Scharf C, Hecker M. 2003. Global characterization of disulfide stress in *Bacillus subtilis*. J Bacteriol 185:1967–1975 and Walgraeve et al., see above), we decided to determine absolute protein abundances one hour after stress induction of the diamide stress. This has now been mentioned in lines 246-249. Of note, this particular condition has, so far, not been investigated in great detail.

Author's response to the comments of Reviewer #3:

Ferrero-Bordero et al. report a proteomics study of the disulfide stress response in *B. subtilis*. This work provides an important resource for the community of researchers interested in *B. subtilis* physiology as well as researchers interested in disulfide stress responses. The strength of the study is that it builds upon the strong prior work from this group, the proteome results are quantitative and absolute values, and the dataset includes both the membrane and exoproteome. The paper is very clear and well-written, but there are a number of places where the data could use a bit more explanation.

Response: We thank the Reviewer for the constructive comments and suggestions, which have helped us to improve our manuscript..

Major comments.

The difference in optical density between the exponential phase and stationary phase cells is only ~10% (line 165), yet the difference in OD(500) is 3-fold (from 0.4 to 1.2; line 550). Can the authors explain this discrepancy?

Response: We thank the reviewer for pointing out this issue which had unfortunately escaped our attention. Although we had quantified the cell numbers during our experiments, cell numbers from a previous manuscript of our team were used to estimate the numbers of proteins per cell (Maaß, S, Wachlin, G, Bernhardt, J, Eymann, C, Fromion, V, Riedel, K, et al. 2014. Highly precise quantification of protein molecules per cell during stress and starvation responses in *Bacillus subtilis*. Mol Cell Proteomics 13, 2260–2276). These bacterial cell numbers were determined under the same growth conditions as the growth conditions implemented in our present study (Belistky minimal medium, 37 °C, and sampling at the same timepoints). Accordingly, we assumed that the data would be comparable. Nonetheless, as the reviewer points out correctly, there is a discrepancy between the published cell numbers from our previous study and the data obtained from our present experiments. Indeed, the determined cell number in the presently described experiment matches the 3-fold increase in the OD₅₀₀. Hence, we have recalculated the absolute protein abundances using the cell counts determined during our present experiments. The text (lines 182-198), Supplemental Table 2, and Figures 2 and S2 were revised accordingly.

2. L. 169. Can the authors reassure the reader that this estimate of number of translocons per cell is valid under their growth conditions? Since at least some proteins are translocated co-translationally, does this flux (approx. 2 min. on average for a protein to transit the membrane) make biological sense? It seems reasonable if proteins are translated at ~20 aa/second and one assumes translocation may (on average) be slower than translation, or may stall, or may involve only a subset of translocons.

Response: The presently used Sec translocon numbers were obtained in our previous study (Antelo-Varela, M, Bartel, J, Quesada-Ganuzo, A, Appel, K, Bernal-Cabas, M, Sura, T, et al. (2019). Ariadne's thread in the analytical labyrinth of membrane proteins: integration of targeted and shotgun proteomics for global absolute quantification of membrane proteins. Anal. Chem. 91, 11972–11980), where exactly the same cultivation conditions were used as in our present study. We therefore believe that the mentioned estimate of the number of Sec translocons per cell is valid also in our present study,

despite some growth differences as discussed in our response to the previous comment of the reviewer.

3. LI 170-171. The authors refer in several places to the subset (a minority!) of the exoproteome that is predicted to transit through the Sec system. Can this annotation please be added to the data files? As presented, I did not see a column to indicate which proteins are predicted to contain signal sequences (is this the column "GP4 cellular location" in one of the SI tables?).

Response: We thank the reviewer for the suggestion to add more information to the Supplemental Material. Two columns have been added in the Supplemental Tables that list the exoproteome abundances (Supplemental Tables 2 and 4). The first added column lists the protein localizations predicted with the GP4 tool, and the second added column lists the predicted export system used to secrete the respective proteins. The latter column also refers to non-canonical secretion pathways and predicted protein interactions with the bacterial cell wall.

Notably, for our calculations on Sec-dependent protein secretion, only abundances of proteins predicted to be secreted through the Sec pathway were included (lines 178-198).

4. Please label the SI tables so they are numbered (as downloaded from the website it was hard to identify which was which).

Response: Table names and captions have been added to each Table and in each Tab. The file names represent the Table names and numbers. Moreover, we have taken care to appropriately label the Tables during the submission process.

5. L. 176. At this point in the text, perhaps clarify for the reader that this high fraction of metabolic enzymes is due (I think?) to the presence of many enzymes associated with cytosolic metabolic pathways in the exoproteome.

Response: We thank the reviewer for the suggestion and agree that it should be emphasized that a lot of extracellularly identified enzymes allow *B. subtilis* to degrade and utilize a wide range of nutrients from the environment to fuel its metabolism. We have revised the text accordingly (lines 203-213). However, in order to not expand the manuscript too much, we have decided to remove the paragraph on secreted stress proteins (lines 214). We hope for the reviewer's agreement.

6. L. 220. Is it possible to convert these numbers into a fractional occupancy of the membrane surface? The number I have heard is close to 50%, but I would be interested in the authors' thoughts on this point.

Response: Although the fractional occupancy of the membrane surface by proteins is a very interesting feature to determine, it is difficult to do this with our current dataset. We have searched the literature for formulas to calculate the fractional area of the membrane occupied by proteins and found two publications on this topic (see below). However, to our understanding, we would also need information on the lipid composition of the cells under the applied experimental conditions. As the lipid composition was not determined in our present study and, to our opinion, should not be inferred from available literature, we prefer to not present any results of such calculations in our revised manuscript in order to keep speculations to the minimum. We hope for the reviewer's understanding.

Dupuy AD and Engelman DM. 2008. Protein area occupancy at the center of the red blood cell membrane. *Proc Natl Acad Sci U S A* 105, 2848–2852.

Cliff L, Chadda R, and Robertson JL. 2020. Occupancy distributions of membrane proteins in heterogeneous liposome populations. *Biochim Biophys Acta Biomembr* 1862, 183033.

7. L. 252. Is it possible to indicate the regulators involved? For example, which of these proteins are in the Spx regulon?

Response: We thank the reviewer for this useful suggestion. We have now specified the regulators known to be involved in the expression of the different proteins in all Tabs of Supplementary Table 3. The respective information on regulators was retrieved from the Subtiwiki website.

8. L. 299. The basis for induction of the *cadA* gene by diamide has been described (see Fig. 5 in PMID 25213752). Repression of *cadA* requires the CzrA transcription factor that senses cellular Zn(II) levels. At the levels of DIA the authors use (1 mM), thiol oxidation triggers an increase in cytosolic Zn from both the buffered BSH pools and ribosome-associated, Zn-containing r-proteins.

9. Related mechanisms involving perturbations of cellular metal pools may account for the effects on other metal homeostasis systems, or in some cases the regulator may itself be oxidized. For example, it has been suggested that PerR may be oxidized by DIA, but in vitro studies suggest that this reaction requires high DIA levels and protein denaturants! (PMID 16766519).

Response: We thank the reviewer for the two comments 8 and 9 and the respective references. We have incorporated the mentioned aspects on *cadA* regulation in our revised manuscript (lines 358-365).

10. L. 313 and Figure 4. Yes, and both are members of the SigW regulon, so a pathway for induction can be postulated. This may explain why FabF is selectively upregulated as highlighted in Fig. 4.

Response: The reviewer's useful suggestion has now been addressed in the Discussion section of our revised manuscript (lines 586-607).

11. L. 471. I agree! The non-canonical secretion of cytosolic proteins to the exoproteome is a major and long-standing puzzle in proteomics. This point is, unfortunately, not as widely appreciated as it should be. I appreciate that the authors have cited the TIM review by Götz.

Response: We thank the reviewer for this positive feedback. We fully agree that the information on non-canonical protein secretion, which we have now included in the Supplemental Tables 2 and 4 (see point 3), will be very useful for researchers interested in the full spectrum of protein export mechanisms.

12. L. 899. Please define the difference between active and standby secretion.

Response: The median protein secretion rate was calculated to be 1.8 molecules per minute for extracellular proteins. Hence, in Figure 3C, we defined proteins with a higher secretion rate than the median rate (e.g., TasA with a secretion rate of 3.3 molecules per minute) to be efficiently secreted. In our original manuscript we referred here to "active secretion", but this may be confusing for the reader, because proteins that are exported from the cytoplasm via the Sec pathway are actively secreted by definition. Proteins with a secretion rate lower than the median protein secretion rate of 1.8 molecules per minute were classified to exhibit "standby secretion". We have included this information in the caption of Figure 3C (lines 1030-1033).

Minor comments

I. 35, 707, 744 typos.

II. 138-139: please define abbreviations when first introduced.

I. 141. To prove (not to proof)

Response: The typos have been corrected and the text has been revised in accordance with the reviewer's suggestions.

Re: Spectrum02616-23R1 (From the outer space to the inner cell: Deconvoluting the complexity of *Bacillus subtilis* disulfide stress responses by redox state and absolute abundance quantification of extracellular, membrane and cytosolic proteins)

Dear Dr. Sandra Maaß:

Although one of the reviewers had a couple of minor suggestions, I see that all the reviewers questions/suggestions have been taken into account with the revised manuscript. They all have no concerns in accepting the revised manuscript for publication.

Your manuscript has been accepted, and I am forwarding it to the ASM production staff for publication. Your paper will first be checked to make sure all elements meet the technical requirements. ASM staff will contact you if anything needs to be revised before copyediting and production can begin. Otherwise, you will be notified when your proofs are ready to be viewed.

Sincerely,
Ryan Rego
Editor
Microbiology Spectrum

Reviewer #2 (Comments for the Author):

Comments were addressed.

Reviewer #3 (Comments for the Author):

Overall, the authors have done an excellent job of addressing the concerns noted by the referees. However, a couple of aspects of the presentation could use further explanation or clarification.

1. I agree with the authors that it is very desirable to gain a holistic understanding of protein secretion (l. 176 in marked text). Their calculations suggest that protein secretion is occurring at a rate of one protein exported every ~2 minutes (34 per hr) through 56 translocons per cell. Overall, they suggest ~32 molecules exported per cell per minute. Appropriately, this is based only on the subset of proteins with signal peptides. Do the authors wish to note that the actual number of translocons available

for exoproteins is likely less than 56 per cell if a fraction are engaged in the insertion of membrane proteins? Do they have a sense of how significant this correction might be? Since translation rates are on the order of 10 aa per second, a 600 aa protein that is co-translationally exported would take about a minute, so the estimates above do seem reasonable.

2. I. 202 "a prominent portion of secreted proteins can be assigned to metabolic processes..." I believe that this statement refers to the total exoproteome and not the subset that are secreted through the Sec system. Is it appropriate to call these "secreted proteins" when the mechanism by which they reach the medium is not known (possibly lysis, but other non-conventional secretion pathways have been suggested).